# LAVA: A unified framework for finetuning language and vision models

**Daorui Ding** [1]   **Fanhua Shang** [1]   **Tiancan Feng** [1,2]   **Junkang Liu** [1]   **Hongying Liu** [3]

## Abstract

LoRA and its variants have attracted considerable attention because of their abilities to tune a negligible number of parameters while achieving comparable downstream performance. This success is largely attributed to the intrinsic low-rank structure of model parameter spaces, which allows LoRA to train two projection matrices to project weights into a low-dimensional subspace and then map them back. However, LoRA tends to use more subspace for optimization while under-utilizes its effective degrees of freedom. Moreover, when using LoRA to tune convolution layers, a flatten operation is required to convert tensors into matrices. We argue that this will degrade the model's performance. In this paper, we propose a unified **L**anguage **A**nd **V**ision **A**daption finetuning framework (called **LAVA**). Specifically, we verify the existence of low-rank subspaces in convolution layers empirically and present to parameterize the increment of both convolution kernels and matrices as sum of learnable rank-1 components. To improve training stability, we analyze the optimization dynamics of LoRA and incorporate orthogonal regularization into our parameterization, for which we give theoretical proof that it will help reduce the variance of the gradient. We conduct various experiments on different downstreaming tasks to validate LAVA's superiority. For example, when tuning LLaMA2-7b for commonsense tasks, the performance of our LAVA is **+1.9%** higher than that of LoRA. For metric depth estimation tasks, LAVA only tunes $\sim$1.5% of Depth-Anything$_{\text{large}}$ (335.3M), and achieves **+3.5%** $\delta_1$ accuracy against that of LoRA and **+5.6%** $\delta_1$ accuracy against that of SVDiff.

## 1. Introduction

Pre-trained large models (PLMs) have shown their remarkable abilities across wide domains (Wang et al., 2025) (Siméoni et al., 2025) (Guo et al., 2025). By scaling the number of parameters and size of data, models exhibit unprecedented generalization (Chen et al., 2026) abilities. To tailor these models into specific task domains and bypass the bottleneck of the accelerator's memory, parameter-efficient fine-tuning methods (PEFT) are proposed. Supported by intrinsic dimensionality theory that there exists a low-dimensional subspace that is important for downstream tasks (Aghajanyan et al., 2021), LoRA freezes the original weight matrix and tunes two newly introduced matrices in this low-rank space to significantly reduce memory cost.

However, we argue that LoRA and its variants fail to explore the low-rank subspace sufficiently. The optimization of the low-rank factors $A$ and $B$ is typically unconstrained. As a result, multiple columns of $A$ (or rows of $B$) can collapse into highly correlated directions: we observe that LoRA tends to use the entire low-rank subspaces for training, while the effective rank is much smaller.

Additionally, in some vision tasks that require pixel-wise granularity (e.g., image generation (Podell et al., 2024), depth estimation (Bhat et al., 2023), and image inpainting (Ju et al., 2024)), finetuning convolution layers is still needed: it can refine the regional details of images. Researchers normally full fine-tune the convolution layer, which ignores the existence of intrinsic dimensions in the convolution kernel. On the other hand, existing PEFT methods focus on attention and neglect the performance tuning on convolution. A naive way is to flatten the convolution weights into matrices, use LoRA to finetune them, and then restore them to the original tensor dimension. However, such a reshape-tune-restore training paradigm will inevitably force dimension disorder and disrupt the spatial encoding properties inherent in convolution. Inspired by the analysis above, we naturally propose a question:

> ***Can we design a unified PEFT framework that adapts both attention- and convolution-based modules efficiently in low-rank subspaces across NLP and vision tasks?***

In this paper, we propose **LAVA**, a unified **L**anguage **A**nd **V**ision **A**daptation framework. The central idea is to think of

[1]School of Computer Science and Technology, Tianjin University, Tianjin, China [2]Zhonguancun Academy [3]Medical College, Tianjin University, Tianjin, China. Correspondence to: Fanhua Shang <fhshang@tju.edu.cn>, Hongying Liu <hyliu2009@tju.edu.cn>.

*Proceedings of the 43$^{rd}$ International Conference on Machine Learning*, Seoul, South Korea. PMLR 306, 2026. Copyright 2026 by the author(s).

matrices as 2D tensors and introduce generalized subspaces that keep dimensional integrity. We leverage dimension-isolated matrices and parameterize the incremental update into the sum of learnable rank-1 components. We show that when LAVA is applied to attention, it is a general form of LoRA. Moreover, to reduce dimension redundancy, we propose column orthogonal regularization and give theoretical proof that this helps reduce the variance of the gradient and stabilize the training. We validate LAVA across a wide range of tasks in various model architectures, and we evaluate our method in different model scales (from 125M in RoBERTa up to 7B in LLaMA2-7b). The experiment results show that LAVA consistently outperforms other PEFT methods across every domain: for example, in commonsense reasoning tasks, LAVA is on average **+1.9%** higher than that of LoRA; and in depth estimation, the performance of tuning LAVA on Depth-Anything is **+3.5%** higher than that of LoRA.

Our main contributions can be summarized as follows:

- We propose a hypothesis that low-rank subspaces exist in convolution layers, and we empirically validate the existence of such subspaces.

- Based on the observation above, we propose LAVA, a unified language and vision adaptation method that introduces a general framework for tuning convolution and attention blocks.

- We theoretically prove that our proposed orthogonal regularization can stabilize the training of LAVA by reducing the variance of the gradient, and we empirically validate the robustness of different choices of hyperparameters and give insights on how to set hyperparameters accordingly.

- We conduct extensive experiments to show that LAVA consistently surpasses other PEFT methods across various tasks and models: from LLaMA2-7b in commonsense reasoning to stable-diffusion-XL in text-to-image generation.

## 2. Related Works

We first give the preliminaries used throughout this paper. A matrix is denoted by an upper-case letter, e.g., $A$; $A_{ij}$ denotes the element of $A$ at the $i$-th row and $j$-th column. An $N$-th order tensor is denoted by a calligraphic letter, e.g., $\mathcal{X} \in \mathbb{R}^{I_1 \times I_2 \times \cdots \times I_N}$, where $N$ is the number of dimensions of the tensor, and we use python-style $A_{[:,r]}$ to slice the entire column $r$ of matrix $A$.

### 2.1. Low-Rank Adaptation (LoRA)

LoRA (Hu et al., 2022) is a common method in parameter-efficient finetuning. It uses low-rank matrices to approximate the real weight changes. When training, LoRA freezes

the pre-trained weight matrix $W \in \mathbb{R}^{n \times m}$, and models the weight update $\Delta \in \mathbb{R}^{n \times m}$ with the multiplication of two smaller trainable matrices $A \in \mathbb{R}^{r \times m}$, $B \in \mathbb{R}^{n \times r}$, where $r \ll \min\{n, m\}$. We denote the input token of the current layer as $x \in \mathbb{R}^D$, the output as $y \in \mathbb{R}^D$, where $D$ is the hidden dimension of the model. Assuming $n = m = D$, the vanilla feed-forward pass is formulated as follows:

$$y = (W + BA)x. \tag{1}$$

The gradients with respect to $A$ and $B$ are:

$$\begin{cases} \nabla_A L = B^\top \dfrac{\partial L}{\partial y} x^\top = B^\top (\nabla_W L), \\ \nabla_B L = \dfrac{\partial L}{\partial y} x^\top A^\top = (\nabla_W L) A^\top, \end{cases} \tag{2}$$

where $L$ is the next-token prediction loss.

Recently, many variants of LoRA have been proposed to improve its performance from different perspectives. Inspired by weight normalization (Salimans & Kingma, 2016), DoRA (Liu et al., 2024) separates the direction and magnitude of the adapted matrix $BA$ apart: it introduces column-wise normalization on $BA$ so that it only controls the optimization direction, and a vector $m$ is trained to control the scale of each column. (Hayou et al., 2024) analyzes the infinite-width setting and finds that A and B are not properly trained, and the asymmetry between them leads to inefficient feature learning. Thus, they propose to let the learning rate of the matrix $B$ be $\lambda$ times larger than that of $A$ to learn features sufficiently. Conv-Adapter (Chen et al., 2024) is a PEFT method designed for tuning convolution layers. It trains two smaller convolution layers, where one for aligning the size of the feature map with the output generated after the pre-trained convolution layer, and the other is a 1x1 convolution layer designed for controlling channel depth.

### 2.2. Comparisons against existing methods

In this section, we provide a detailed comparison between LAVA and existing methods to properly situate our work within the literature. There are some works about introducing orthogonal matrices into PEFT: HRA (Yuan et al., 2024) bridges low-rank and orthogonal adaptation using householder reflections, and SORSA (Cao, 2024) combines SVD-based adaptation with orthonormal regularization and analyzes its effect on conditioning. However, LAVA incorporates orthogonality into a low-rank tensor-form adaptation to preserve the simplicity of additive PEFT. Another line is for tensor-form PEFT methods. Recently, several works have found the weakness of finetuning in matrix form and proposed tensorized finetuning methods: ReFTA (Zheng et al., 2026) argues that existing PEFT methods are limited by layer-wise low-rank structure and proposes to stack mod-

ules and finetune them in a tensor form, and TLoRA (Tao et al., 2025) uses tensor-train matrix to form rotation matrix, and tensor-ring decomposition for the residual. ReFTA focuses on using the relevance across layers, and TLoRA studies the low-rank approximation gap. Instead, LAVA focuses on reshape-free tensor-form update for convolutional kernels, while still supporting original matrix tuning within a unified framework. Lastly, there are some works on finetuning convolution layers as well: Conv-LoRA (Zhong et al., 2024) injects convolutional modules into LoRA to adapt Segment Anything model; Conv-Adapter (Chen et al., 2024) studies efficient transfer for ConvNets using lightweight adapter modules. In contrast, LAVA does not add extra convolutional branches or adapter blocks and can be applied beyond models from the vision fields.

## 3. Low-rank subspace analysis

Drawing inspiration from Candecomp/Parafac (CP) decomposition, which represents the tensor as a sum of rank-1 tensor components while keeping the relations of the tensor dimensions, we introduce a new tensor increment analysis. Our analysis compares finetuning methods with and without reshaping operations and validates the existence of low-rank subspaces for convolution weights.

**Analysis Method**: Denote the weight of a convolution layer as $W_{tensor} \in \mathbb{R}^{c_{out} \times c_{in} \times h \times w}$, we compare two finetuning methods: reshape-involved (LoRA) and reshape-free. For reshape-involved methods, we reshape the original tensor into the shape $W' \in \mathbb{R}^{c_{out} \times (h \times c_{in} \times w)}$ and then use LoRA to tune the flattened matrix. On the other hand, reshape-free method parameterizes the update $\Delta W \in \mathbb{R}^{c_{out} \times c_{in} \times h \times w}$ into sum of rank-1 tensors. It keeps the dimensional integrity by introducing one matrix per tensor dimension. Suppose the rank of reshape-free way is $R$ and $X \in \mathbb{R}^{h \times R}$, $U \in \mathbb{R}^{c_{out} \times R}$, $Y \in \mathbb{R}^{w \times R}$ and $V \in \mathbb{R}^{c_{in} \times R}$, it directly reparameterizes the update tensor as the sum of $R$ outer product of four vectors:

$$\Delta W = \sum_{r=1}^{R} U_{[:,r]} \circ V_{[:,r]} \circ X_{[:,r]} \circ Y_{[:,r]}, \quad (3)$$

where $\circ$ denotes the outer-product operation. we conduct a toy experiment to compare the performances on ResNet18 (He et al., 2016), ResNet34 (He et al., 2016), VGG11 (Simonyan & Zisserman, 2015), AlexNet (Krizhevsky et al., 2012), GoogLeNet (Szegedy et al., 2015), and ConvNeXt (Liu et al., 2022) between methods with and without keeping dimensional integrity in low-rank subspaces. We inject trainable parameters into the convolution layer and finetune the model on the dataset CIFAR10 (Krizhevsky et al., 2009).

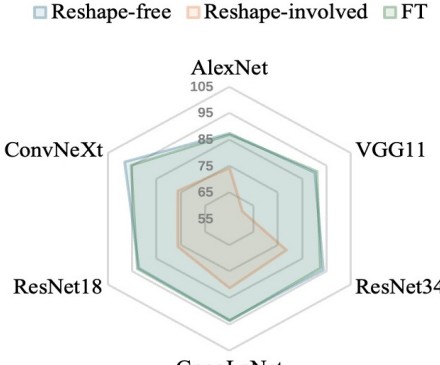

**Classification Accuracy**

☐ Reshape-free ☐ Reshape-involved ☐ FT

*Figure 1.* Comparison of the accuracies of fine-tuning among reshape-free, reshape-involved, and full-finetuning methods.

*Table 1.* Trainable parameters of reshape-involved and reshape-free methods on six pre-trained models.

| Model | Reshape-involved | Reshape-free |
|---|---|---|
| AlexNet | 0.13 | 0.08 |
| VGG11 | 0.09 | 0.04 |
| ResNet34 | 1.05 | 0.25 |
| GoogLeNet | 2.32 | 1.66 |
| ResNet18 | 1.00 | 0.28 |
| ConvNeXt | 3.12 | 0.15 |

As is shown in Fig. 1 and Table 1, the reshape-free method outperforms that of reshape-involved by around 10% in all models using less than ≈50% trainable parameters. What's more, models finetuned by the reshape-free method even outperform full finetuning in ConvNeXt and VGG11, while the reshape-free method only tunes only ~0.1% of total parameters. Detailed quantitative results can be found in Appendix A.5. To make our claim clearer, we provide comparisons across different unfolding choices in Table. 11 in the Appendix so that the discussion is not tied to one specific reshape order (all variants are compared under the same hyperparameters). Based on the observation, we claim that:

> *when a convolutional kernel is first unfolded into a matrix and then finetuned by a low-rank update, the parameterization becomes unfolding-dependent and no longer explicitly preserves the kernel's original multi-way structure.*

The performance superiority from toy example gives us two insights:

- **Insight 1**: **It is important to preserve the dimension of tensors when finetuning the convolution layer**:

maintaining the original multi-dimensional form ensures spatial correlations;

- **Insight 2**: **Convolution layers also exhibit intrinsic dimensionality during finetuning**: although convolution kernels are high-dimensional, only a small subset of directions in this space contribute effectively to downstream adaptation.

# 4. LAVA: Generalized finetuning framework for both vision and language tasks

Drawing from the insights of our subspace analysis, in this section, we introduce a unified framework for finetuning language and vision models. Our method contains two important components: (i) low-rank subspace-based adaptation, which parameterizes the increment into a sum of rank-1 tensors; (ii) orthogonal regularization, which is applied on trainable blocks to stabilize the training.

## 4.1. Reducing Tensor to Matrix: Generalized subspace-based Adaptation

When updating convolution or higher-order tensors, we propose to use Eq. (3) to reparameterize the increment. This structure intrinsically isolates gradient computations and parameter updates per tensor dimension, as the partial derivatives $\frac{\partial L}{\partial (U_{[:,i]} \circ V_{[:,i]} \circ X_{[:,i]} \circ Y_{[:,i]})}$ depends solely on vectors related to $i$-th column. By representing $\Delta W$ as a sum of separable components, the training of $X$, $Y$, $U$, and $V$ becomes dimension-wise independent, reducing cross-dimensional interference.

As a matrix can be thought of as a 2D tensor, LAVA can be used in attention blocks as a general method of LoRA. For one matrix, the tensor dimension reduces to 2, and Eq. (3) can be reformulated into the following form:

$$W = W_0 + \Delta \approx W_0 + \sum_{r=1}^{R} U_{[:,r]} \circ V_{[:,r]}$$
$$= W_0 + \sum_{r=1}^{R} U_{[:,r]}(V_{[:,r]})^\top = W_0 + UV^\top, \quad (4)$$

where $W \in \mathbb{R}^{m \times n}$, $U \in \mathbb{R}^{m \times R}$ and $V \in \mathbb{R}^{n \times R}$. Here, each $U_{[:,r]}(V_{[:,r]})^T$ is a rank-1 residual matrix to compensate for the gap between the expected and real incremental. And we notice that Eq. (4) exactly matches the form of LoRA, meaning that LoRA is a special case of LAVA.

For the original tensor $\mathcal{X} \in \mathbb{R}^{c_{out} \times c_{in} \times h \times w}$, the memory required to store the tensor is $\mathcal{O}(c_{out} \times c_{in} \times w \times h)$. Assuming the rank is set to $r$, in reshape-involved LoRA, the memory is $\mathcal{O}((c_{out} + c_{in} \times h \times w) \times r)$, while in our method, the memory is $\mathcal{O}((c_{out} + c_{in} + w + h) \times r)$, which is typically far lower than that of reshape-involved LoRA.

## 4.2. Extending Matrix to Tensor: Orthogonal Regularization Helps Stabilize The Training

Consider LAVA in matrix form: $\Delta \approx \sum_{r=1}^{R} U_{[:,r]}(V_{[:,r]})^\top$, each $V_{[:,r]}$ can be represented as a basis vector in the subspace $\mathcal{S} \subset \mathbb{R}^R$ and the corresponding $U_{[:,r]}$ determines the coordinate along this dimension. Unconstrained optimization may lead to an ill-conditioned subspace, whereas the orthogonal regularization in LAVA improves conditioning and can suppress redundant directions. We propose to apply column-orthogonal regularization $\lambda\|AA^\top - I\|_F^2$ on $A$ during optimization to enforce orthogonality. In this situation, the gradient of $B$ keeps the same, and the gradient of $A$ becomes:

$$\nabla_A L = B^\top(\nabla_W L) + 4\lambda A(A^\top A - I). \quad (5)$$

The proposed orthogonal regularization indeed helps stabilize the training and reduce the variance in gradients. In deed, in LLM finetuning tasks, learning rate is often set to be small, thus the discrete optimization process can be interpreted as a numerical integration of a continuous-time dynamical system, and the optimization process can be modeled using gradient flow. Considering the residual part $D = A^\top A - I$ and its first term $C = A^\top A$, let $H = A^\top B^\top G + G^\top BA$ and $G = \nabla_W L$, then they can be modeled using the following directives:

$$\begin{cases} \frac{dA}{dt} = -B^\top G - 4\lambda A(A^\top A - I), \\ \frac{dB}{dt} = -GA^\top, \\ \frac{dC}{dt} = -8\lambda C(C - I) - H, \\ \frac{dD}{dt} = -8\lambda D(D^2 + D) - H. \end{cases} \quad (6)$$

Under this framework, when training LoRA, we can analyze that at any time $t$, the Frobenius Norm $\|A(t)\|_F$ and $\|B(t)\|_F$ is upper bounded.

**Theorem 4.1** (Upper bound of Frobenius Norm $\|A(t)\|_F$ and $\|B(t)\|_F$). *If the optimizer is SGD, denote the full gradient as $\bar{G}(t)$, and the gradient calculated on a minibatch as $G(t)$. Suppose $\mathbb{E}[G(t)] = \bar{G}(t)$, $\|\bar{G}(t)\|_2 \leq \mu$, and $\mathbb{E}[\|G(t) - \bar{G}(t)\|_2^2] \leq \sigma^2$. Denote $v(t) = \|A(t)\|_F^2 + \|B(t)\|_F^2$, $v_0 = \|A_0\|_F^2 + \|B_0\|_F^2$, and $\gamma = \frac{\mu^2 + \sigma^2}{2\lambda}$. Then, for any finite training horizon $T$, we have*

$$\sup_{t \in [0,T]} \mathbb{E}[v(t)] \leq v_0 e^{\gamma T} + \frac{16\lambda^2 m}{\mu^2 + \sigma^2}\left(e^{\gamma T} - 1\right). \quad (7)$$

The proof of Theorem 4.1 can be found in Appendix A.2. This theorem indicates that the energy $v(t) = \|A(t)\|_F^2 + \|B(t)\|_F^2$ cannot grow without bound. Equivalently, When the factors drift to large norms, the regularization-driven component in $\frac{dA}{dt}$ provides a restoring force that counteracts this drift. Thus, orthogonal regularization acts as a friction term that prevents runaway trajectories and confines optimization to a bounded region in parameter space.

**Theorem 4.2** (Convergence rate under orthogonal regularization). *If the optimizer is SGD, when the regularization strength term is set to $\lambda$, $A$ will get converged to orthogonal matrix $M$ at a speed of $\mathcal{O}(\frac{\mu+\sigma}{\lambda})$.*

It characterizes the convergence behavior of matrix $A$ towards an orthogonal structure, thereby offering a quantitative measure of how quickly the orthogonal term diminishes during optimization. Please refer to Appendix A.3 for the proof. Based on the theorem above, we can get the following proposition:

**Theorem 4.3** (Upper bound of gradient variance). *Let $G = \nabla_W L \in \mathbb{R}^{m \times n}$, define the covariance matrix as $\Sigma_G = \mathbb{E}[(G - \mathbb{E}G)^\top (G - \mathbb{E}G)] \in \mathbb{R}^{n \times n}$, deviation error as $E_A = \|A^\top A - I\|_2$. Suppose $tr(G) < +\infty$, then:*

$$\mathrm{Var}(\nabla_B L) \leq (1 + E_A) tr(\Sigma_G).$$

As $E_A$ converges to 0 at the speed of $\mathcal{O}(\frac{\mu+\sigma}{\lambda})$, the coefficient term $(1 + E_A)$ will get closer to 1, and the variance of $\nabla_B L$ will fall into a small neighborhood region of *FT* variance $\Sigma_G$. In other words, penalizing $A$ with orthogonal regularization will reduce the variance of the gradient in $B$, and the algorithm is more likely to converge to a point with smaller gradient and a more stable oscillation radius. The detailed proof can be found in Appendix A.4.

Such regularization can be extended to tensor form as well. Suppose a 2D convolution layer $\mathcal{X} \in \mathbb{R}^{c_{out} \times c_{in} \times h \times w}$, LAVA decomposes the update of $\mathcal{X}$ into Eq. (3). We apply column-orthogonal regularization on $V$. Since LAVA decomposes the incremental into the sum of rank-1 tensors, adding orthogonal regularization will help LAVA to explore the rank-$R$ subspace more sufficiently while obeying dimensional integrity.

# 5. Experiments and Results

In this section, we evaluate how orthogonal regularization benefits the training in attention blocks in NLP tasks. Then we switch to vision and multimodal tasks where convolution layers exist: we test whether LAVA is applicable for both convolution and matrix.

## 5.1. Natural Language Understanding

Firstly, we test LAVA on Natural Language Understanding (NLU) to showcase the effectiveness of applying orthogonal regularization in NLP tasks.

We finetune RoBERTa_base (125.0M) on GLUE benchmark (Wang et al., 2018) following (Wu et al., 2024a). In this experiment, we compare our method with baselines including Full-Finetuning, BitFit (Ben Zaken et al., 2022), Adapter (Houlsby et al., 2019), Adapter-FFN (Pfeiffer et al.,

2021), LoRA (Hu et al., 2022), RED (Wu et al., 2024a), LoReFT (Wu et al., 2024b), and DeLoRA (Bini et al., 2025). Please refer to Appendix A.6 for detailed descriptions of the datasets, baseline, and hyperparameters.

**Main results.** Table 2 shows experiment results on the GLUE benchmark: our LAVA method outperforms all the baseline methods and achieves the highest average score (85.9%), **0.3%** higher than DeLoRA and **1.2%** higher than LoRA. Additionally, for MRPC, CoLA, QNLI, and STS-B, our method consistently outperforms FT. For MRPC, our method even achieves superior accuracy over FT by **2.2%**. Compared with other PEFT methods, LAVA beats all other methods except on CoLA and RTE. But on these two datasets, performances are heavily influenced by random seeds, and we argue that the under-performances are not representative. The experimental results suggest that the integration of orthogonal regularization could help explore the low-rank spaces better and enhance learning capability.

## 5.2. Commonsense Reasoning

Then we scale the size of the model up to Gemma2-2b, LLaMA3-3b and even to LLaMA2-7b (Touvron et al., 2023) and turn to commonsense reasoning tasks to validate LAVA's compatibility in large models at a larger scale.

In this experiment, we evaluate LAVA against LoRA and include ChatGPT's accuracy obtained with gpt-3.5-turbo API using a zero-shot Chain of Thought (Wei et al., 2022). For fair comparison, we set the rank of all the finetuning methods as 32.

From the results in Table 3, we observe that orthogonal regularization in LAVA significantly improves LoRA's learning ability and enhances LoRA's performance, achieving +1.9% in LLaMA2-7b, even far out-performing the closed-resource model GPT-3.5-turbo. This shows that our LAVA method can be scaled to large models as well, demonstrating its generality across LLMs with different sizes. Detailed results on all the datasets are shown in Appendix A.7.

Next, we evaluate LAVA's versatility and switch to models in vision and multimodal because these models normally have convolution to extract regional information and attention for global modeling.

## 5.3. Semantic Segmentation

We finetune SAM (Kirillov et al., 2023) using LAVA in three real-world scenarios, medical (Polyp (Bernal et al., 2015) and ISIC2017 (Codella et al., 2018)), natural (camouflaged object detection (Fan et al., 2020) and shadow (Vicente et al., 2016)) and agricultural (leaf disease segmentation (Rath, 2023)) respectively.

We evaluate our method against LoRA (Hu et al., 2022) and Conv-LoRA (Zhong et al., 2024) on all three fields. All the

*Table 2.* Comparisons of different methods finetuning RoBERTa on GLUE benchmark. The best result on each dataset is marked **bold**, and the second highest value is marked underline. Results are averaged on random seeds 42, 43, 44, 45, and 46.

| Method | # Params | MNLI | SST-2 | MRPC | CoLA | QNLI | QQP | RTE | STS-B | Avg. |
|---|---|---|---|---|---|---|---|---|---|---|
| FT | 125.0M | $\mathbf{87.3}_{\pm 0.34}$ | $\mathbf{94.4}_{\pm 0.96}$ | $87.9_{\pm 0.91}$ | $62.4_{\pm 3.29}$ | $92.5_{\pm 0.22}$ | $\mathbf{91.7}_{\pm 0.19}$ | $\underline{78.3}_{\pm 3.20}$ | $90.6_{\pm 0.59}$ | $\underline{85.6}$ |
| Adapter | 0.4M | $87.0_{\pm 0.28}$ | $93.3_{\pm 0.40}$ | $88.4_{\pm 1.54}$ | $60.9_{\pm 3.09}$ | $92.5_{\pm 0.02}$ | $90.5_{\pm 0.08}$ | $69.8_{\pm 1.51}$ | $90.5_{\pm 0.35}$ | 85.0 |
| Adapter-FFN | 0.3M | $87.1_{\pm 0.10}$ | $93.0_{\pm 0.50}$ | $88.8_{\pm 1.38}$ | $58.5_{\pm 1.69}$ | $92.1_{\pm 0.28}$ | $90.2_{\pm 0.07}$ | $77.7_{\pm 1.93}$ | $90.4_{\pm 0.31}$ | 84.7 |
| BitFit | 0.1M | $84.7_{\pm 0.08}$ | $94.0_{\pm 0.87}$ | $88.1_{\pm 1.57}$ | $54.0_{\pm 3.07}$ | $91.0_{\pm 0.05}$ | $87.3_{\pm 0.02}$ | $69.8_{\pm 1.51}$ | $89.5_{\pm 0.35}$ | 82.3 |
| LoReFT | 0.02M | $83.1_{\pm 0.26}$ | $93.4_{\pm 0.64}$ | $\underline{89.2}_{\pm 2.62}$ | $60.4_{\pm 2.60}$ | $91.2_{\pm 0.25}$ | $87.4_{\pm 0.23}$ | $\mathbf{79.0}_{\pm 2.76}$ | $90.0_{\pm 0.29}$ | 84.2 |
| RED | 0.02M | $83.9_{\pm 0.14}$ | $93.9_{\pm 0.31}$ | $\underline{89.2}_{\pm 0.98}$ | $61.0_{\pm 2.96}$ | $90.7_{\pm 0.35}$ | $87.2_{\pm 0.17}$ | $78.0_{\pm 2.06}$ | $90.4_{\pm 0.32}$ | 84.3 |
| LoRA | 0.3M | $86.6_{\pm 0.23}$ | $93.9_{\pm 0.49}$ | $88.7_{\pm 0.76}$ | $59.7_{\pm 4.36}$ | $92.6_{\pm 0.10}$ | $90.4_{\pm 0.08}$ | $75.3_{\pm 2.79}$ | $90.3_{\pm 0.54}$ | 84.7 |
| DeLoRA | 0.3M | $\underline{87.2}_{\pm 0.15}$ | $\underline{94.1}_{\pm 0.70}$ | $89.0_{\pm 0.96}$ | $\mathbf{63.6}_{\pm 1.52}$ | $\underline{92.8}_{\pm 0.51}$ | $90.1_{\pm 0.13}$ | $77.1_{\pm 3.65}$ | $90.9_{\pm 0.31}$ | 85.6 |
| LAVA (ours) | 0.3M | $\mathbf{87.3}_{\pm 0.10}$ | $94.0_{\pm 0.51}$ | $\mathbf{90.1}_{\pm 2.67}$ | $\underline{63.3}_{\pm 2.36}$ | $\mathbf{93.5}_{\pm 0.97}$ | $\underline{90.6}_{\pm 0.07}$ | $77.4_{\pm 2.67}$ | $\mathbf{91.0}_{\pm 0.21}$ | $\mathbf{85.9}$ |

*Table 3.* Comparisons of different PEFT methods on commonsense reasoning task.

| Model | Method | # Params (%) | Avg. |
|---|---|---|---|
| ChatGPT | - | - | 77.0 |
| Gemma2-2b | LoRA | 1.07 | 77.4 |
| | VeRA | 0.02 | 70.2 |
| | LoRA+ | 1.07 | 77.5 |
| | DoRA | 1.09 | 77.3 |
| | LAVA | 1.07 | **78.1** |
| LLaMA3-3b | LoRA | 1.02 | 76.8 |
| | DoRA | 1.03 | 79.5 |
| | LAVA | 1.02 | **79.8** |
| LLaMA2-7b | LoRA | 0.83 | 77.6 |
| | LAVA | 0.83 | **79.5** |

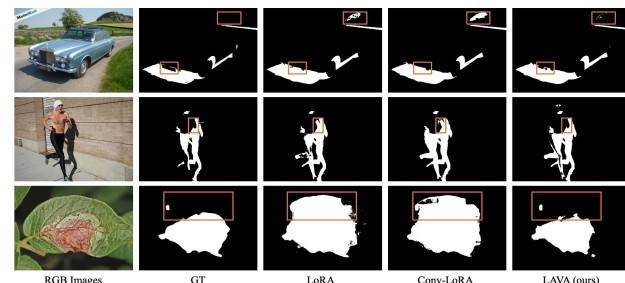

*Figure 2.* Comparisons of LoRA, Conv-LoRA, and LAVA (ours) in binary-class shadow segmentation (**top 2 rows**) and binary-class leaf disease segmentation (**bottom**).

settings follow the experiment configurations from (Zhong et al., 2024). More details of datasets, baselines, and hyperparameters can be found in Appendix A.8.

Fig. 2 shows that LAVA can reinforce image-related local priors, which helps SAM to separate target regions from the background, while LoRA and Conv-LoRA still demonstrate difficulty in separating shadows from black objects. For example, in the second row, these two methods mistakenly recognize the man's arm as the shadow. More quantitative comparisons are shown in Table 19 in Appendix A.8. Here we keep the same metrics as Conv-LoRA (Zhong et al., 2024).

### 5.4. Depth Estimation

Similarly, we scale the proportion of convolution up, and change to depth estimation, where DPT (Ranftl et al., 2021) part (decoder part) of the pre-trained model is made up of convolution blocks. Specifically, in this experiment, we finetune Depth-Anything on nyu-depth v2 (Nathan Silberman & Fergus, 2012). All the experimental details are kept the same with (Yang et al., 2024). Performances are evaluated

using $\delta_1$, $\delta_2$, $\delta_3$, AbsRel, RMSE and log10. More details regarding the baselines are given in Appendix A.9.

**Quantitative comparison.** The results in Table 4 show that our proposed method yields much stronger and accurate depth estimation compared with other finetuning methods. We observe that our method is superior to ZoeDepth, which can be thought of as a previous SOTA method using significantly fewer parameters. Notably, for metric $\delta_1$, our method achieves 0.972, which is 2.1% higher than ZoeDepth. Although there exists a small margin (0.012) with full finetuning Depth Anything, considering the fact that our method only tunes 1.5% of total parameters, such a gap could be accepted and our method still shows potential. It is worth noting that the accuracy of LoRA is less than that of Encoder-only LoRA, meaning that directly using reshape-involved LoRA on convolution layers will degrade the overall performance, which further validates the importance of reshape-free finetuning.

**Qualitative comparison.** Additionally, we provide qualitative comparisons in Fig. 3. Although our method underperforms full-finetuning in metrics, it demonstrates strong robustness in challenging scenarios: From the first and third row, we can observe that LAVA generalizes well in situations with mirrors, which we believe is challenging as

*Table 4.* Comparisons of different methods finetuning Depth-Anything. The best result is marked **bold**. Encoder-only means we only tune the encoder part of the model. † means the value is taken from (Yang et al., 2024).

| Method | params | $\delta_1 \uparrow$ | $\delta_2 \uparrow$ | $\delta_3 \uparrow$ | AbsRel ↓ | RMSE ↓ | log10 ↓ |
|---|---|---|---|---|---|---|---|
| FT (ZoeDepth)† | - | 0.951 | 0.994 | 0.999 | 0.077 | 0.282 | 0.033 |
| FT (Depth-Anything) | 335.3M | 0.984 | 0.998 | 1.000 | 0.056 | 0.206 | 0.024 |
| LoRA (Encoder-only) | 0.79M | 0.941 | 0.992 | 0.998 | 0.079 | 0.346 | 0.108 |
| LoRA | 2.04M | 0.937 | 0.994 | 0.999 | 0.090 | 0.356 | 0.039 |
| SVDiff | 0.11M | 0.916 | 0.991 | 0.998 | 0.102 | 0.399 | 0.044 |
| VeRA | 0.57M | 0.969 | 0.997 | 0.999 | 0.728 | 0.286 | 0.032 |
| Conv-Adapter | 1.21M | 0.969 | 0.997 | 0.999 | 0.072 | 0.278 | 0.031 |
| DoRA | 2.14M | 0.967 | 0.997 | 0.999 | 0.074 | 0.286 | 0.032 |
| LAVA | 1.14M | **0.972** | **0.997** | **0.999** | **0.070** | **0.274** | **0.030** |

the reflection of mirrors will become harder to predict the real depth information. From the second row, we can find that LAVA is robust in scenes with drastic changes in light intensity. These regions are highlighted in orange boxes. More qualitative results are shown in Figs. 7 and 8 in Appendix A.9.

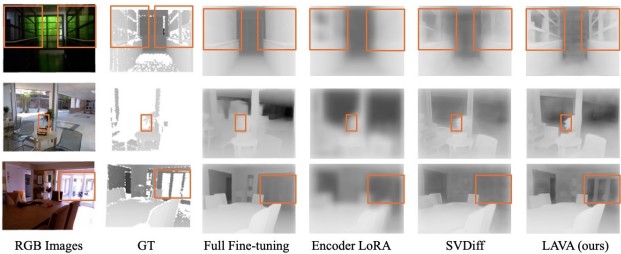

RGB Images    GT    Full Fine-tuning    Encoder LoRA    SVDiff    LAVA (ours)

*Figure 3.* Demonstrations of different finetuning methods on Depth-Anything

### 5.5. Text-to-Image Generation

Finally, we consider using LAVA in finetuning SDXL (Podell et al., 2024), utilizing the training scripts developed by HuggingFace. The target datasets, 3D icons, consist of 23 images, each one showing a company icon in a 3D version with a round-edge square at behind. The random seeds are kept the same for LoRA and our method for fair comparison. We fix the learning rate at 5e-4 and fine-tune the model for 10 epochs following the text-to-image pipeline[1]. Other hyperparameters are set to default values pre-defined in the script. For quantitative comparisons, we use metric FID (Heusel et al., 2017) to compare the similarity of generated images to real ones. The generated images are shown in Fig. 4 and quantitative results are presented in Appendix A.10. More generated images can be found in Fig. 9 in the Appendix.

This result indicates that our method achieves better tuning

---

[1]Refer to `https://github.com/huggingface/diffusers/tree/main/examples/text_to_image`

performances compared with LoRA. For example, in 3d icon datasets, we can observe that for our method, they have clean lines and no redundant details. What's more, they use less saturated colors, which are closer to the Spotify icon. In contrast, images generated by LoRA have sophisticated lines (Columns 1 and 4) and unnatural reflection effect (Column 4), meaning that LAVA could better understand the meaning of the style.

## 6. Ablation study

To isolate the influences of different modules, we conduct an ablation study by finetuning Stable Diffusion XL on the pokemon-blip-captions dataset. In these ablation experiments, all hyperparameters are kept identical except for: (1) the modules to be finetuned, and (2) whether orthogonal regularization is applied. The results are summarized in Table 5: (a) Comparing A1 with A2, and B2 with B3, we can conclude that orthogonal regularization benefits the finetuning of both matrix and convolution forms, as evidenced by a stable increase in CLIP score (an upgrade of 0.14 in attention and 0.02 in convolution). (b) Comparing A1 with B1 indicates that low-rank parameterization is less suitable for unfolded matrices; specifically, when using reshape-involved low rank method to tune convolution layers, the performance undergoes a sharp drop from 29.28 to 28.10. (c) Comparing A1 with B2 reveals that although convolutional layers serve for feature extraction in vision tasks and are typically frozen when well-pretrained, finetuning them still yields a marginal improvement (a CLIP score increase of 0.36). (d) Comparing B1 with B2, the performance gain in B2 (1.52 in CLIP score) strongly validates the necessity of finetuning convolutional layers in their tensor form.

## 7. In-Depth Analysis

Compared with LoRA, our method has an additional hyperparameter: $\lambda$. In LAVA, rank $R$ controls what's the dimension of the subspace, and $\lambda$ controls the convergence rate towards a column-orthogonal matrix. In this section, we analyze different choices of $\lambda$ and rank $R$, and we hope it will give insights into how to tune these two parameters in downstream tasks.

**Robustness of $\lambda$:** We choose $\lambda = 0.1$ as the base, and then test the performances under different $\lambda$ settings on NLU and CIFAR10 classification. For example, the value 2 indicates twice the base, or $\lambda = 0.2$; other values are chosen similarly.

As shown in Figs. 10(a) and 10(b), our analysis shows that LAVA is able to achieve robustness on tuning both attention and convolution across a wide range of $\lambda$ values: ranging from $1/64$ of $\lambda = 0.1$ to $64\times$ of $\lambda = 0.1$. Such a

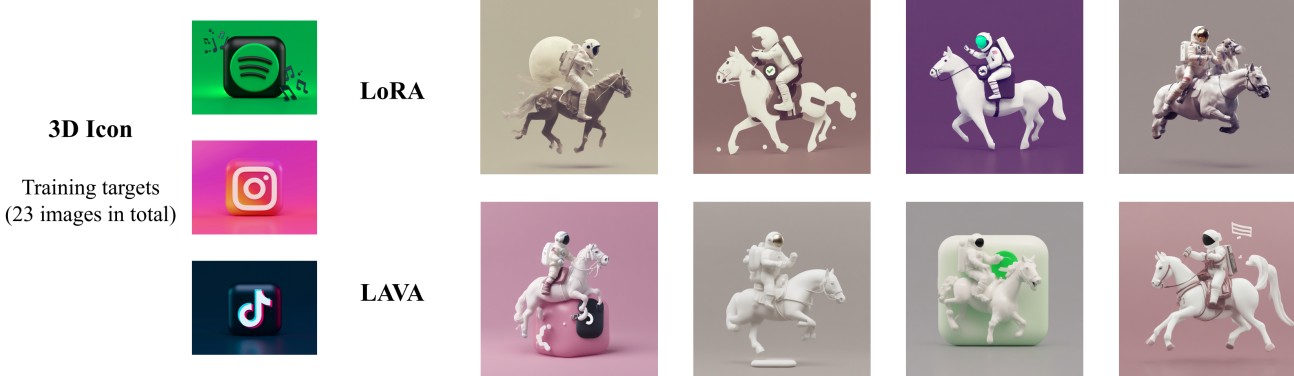

Prompt: a Spotify icon of an astronaut riding a horse, in the style of Spotify.

*Figure 4.* Images generated by SDXL with different finetuning methods on the 3d Icon dataset.

*Table 5.* Ablation study on the effects of different adapter modules and orthogonal regularization during finetuning.

| Setting | ID | Attention Adapter | Conv Adapter | Orthogonal Regularization | Metric ↑ |
|---|---|---|---|---|---|
| Attention-Only | A1 | LoRA | - | False | 29.28 |
| | A2 | LoRA | - | True | 29.42 |
| Full U-Net | B1 | LoRA | Reshape-Involved Low Rank | False | 28.10 |
| | B2 | LoRA | Reshape-Free Decomposition | False | 29.62 |
| | B3 | LoRA | Reshape-Free Decomposition | True (Conv) | 29.64 |
| | B4 | LoRA | Reshape-Free Decomposition | True (both) | 29.96 |

phenomenon could guide us on how to tune $\lambda$ in other downstream tasks: we suggest in NLP tasks, choose a relatively small $\lambda$ value (from $1/32 \times \lambda = 0.1$ to $\lambda = 0.1$; in vision tasks, the range from $1/2 \times$ to $2 \times$ of base $\lambda$ is more preferable.

**Applicable regions of rank $R$:** We further investigate the behaviors of ranks from 3 to 24 and fix orthogonal regularization term to be at $1 \times$ of $\lambda = 0.1$ and test the relevant performances in finetuning CIFAR10 datasets to explore the learning ability of LAVA. This provides us with a quantitative measure of the joint impact of both $\lambda$ and rank $R$. In Fig. 10(c), we show that the classification precisions increase slightly as rank increases. This meets our expectation: each tensor-1 component in LAVA can be recognized as learning from the residual, and the main contribution can be learned from the first few dimensions (there is a performance leap around points at rank 6). Surprisingly, we also observe that, after the main contribution is learned at rank 6, increasing ranks further can still enhance the performance. This means that our LAVA could help alleviate the noise introduced by increasing ranks. We attribute this to our orthogonal regularization, which stabilizes the training and ensures the full exploration in low-rank subspaces.

**How does orthogonal regularization affects the training dynamics?** As discussed in Section 4.2, adding orthogo-

nal regularization can help stabilize the variance of the gradient during training. In this section, we provide experimental results to give intuitive empirical evidence. We compare the variances of gradients on each layer of the model with and without orthogonal regularization. As shown in Fig. 5, when training RoBERTa-base on the QNLI dataset, orthogonal regularization achieves significantly lower gradient variance, which is beneficial to reduce the effect of noise and avoid poor local solutions. More details of the gradient variances of different layers on additional datasets can be found in Fig. 12 in the Appendix.

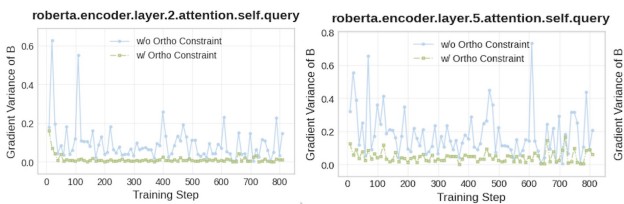

*Figure 5.* Variance analysis of the component $B$ during training.

**What is the additional computational overhead for orthogonal regularization?** We conduct the computational efficiency analysis on the additional cost of orthogonal regularization in training. We provide theoretical computation analysis and memory usage in Table 6. Here, we consider the input as a sequence of tokens, and FF stands for feed-

forward pass, BW: backward pass, LC: loss computation, TP: trainable parameters, CC: computational cost including forward, backward pass, and loss calculation, $T$ stands for the cost of calculating next-token prediction loss, which is widely used in finetuning. We follow the setting $m = n$ here to simplify the analysis. Assuming input $x \in \mathbb{R}^{S \times n}$, $S$ is the sequence length, $n$ is the hidden size of the model, and $r$ is the rank pre-defined before training. Since $r \ll S$ in the long-sequence setting, the additional cost $3rn^2$ is negligible in the training process.

*Table 6.* Computational efficiencies of these PEFT methods.

|  | LoRA / LoRA+ | LAVA |
|---|---|---|
| FF | $y = (W + BA)x$ | $y = (W + BA)x$ |
| BW | $\begin{cases} \nabla_A L = B^\top(\nabla_W L) \\ \nabla_B L = (\nabla_W L)A^\top \end{cases}$ | $\begin{cases} \nabla_A L = B^\top(\nabla_W L) + 4\lambda A(A^\top A - I) \\ \nabla_B L = (\nabla_W L)A^\top \end{cases}$ |
| LC | $\mathcal{L}((W + BA)x, t)$ | $\mathcal{L}((W + BA)x, t) + \lambda \|AA^\top - I\|_F^2$ |
| TP | A, B $(2nr)$ | A, B $(2nr)$ |
| CC | $\mathcal{O}(Sn^2 + 2rn^2 + T)$ | $\mathcal{O}(Sn^2 + 5rn^2 + T)$ |

Besides theoretical analysis, we also compare the wall-clock time spent for forward, backward, and loss calculation phases of 1 epoch. Here we use the time of LoRA as the base, and compare how much longer the time cost is compared to other methods and LoRA.

*Table 7.* Time cost between PEFT methods.

*Table 8.* Speech classification accuracy.

| Method | Conv | Matrix |
|---|---|---|
| LoRA | 1.00 | 1.00 |
| LAVA | 1.20 | 1.18 |
| DoRA | 1.38 | 1.41 |

| Method | Acc (%) |
|---|---|
| LoRA | 98.51 |
| DoRA | 98.56 |
| LAVA | 98.63 |

As shown in Table 7, compared with vanilla LoRA, the overhead of orthogonal regularization can be maintained at not more than 20%. Besides, the overall computation of DoRA is far more than that of vanilla LoRA, making it less applicable in the resource-constrained finetuning phase. In a convolution situation, the overhead of LAVA is still within 20% budget threshold, indicating that LAVA achieves a balance between accuracy and efficiency.

**Is LAVA suitable for other modalities besides vision and languages?** To examine LAVA's generalization beyond the main experimental modalities, we further evaluated it in the speech domain by fine-tuning a pre-trained Wav2Vec2 model with 1B parameters on the SUPERB-KS dataset. As shown in Table 8, LAVA achieves 98.63% accuracy, slightly outperforming LoRA (98.51%) and remaining on par with DoRA (98.56%). These results provide preliminary evidence that LAVA can maintain competitive performance on speech models, suggesting its potential to be further explored across broader modalities in future work.

**What is the difference in ranks between LoRA and LAVA?** In the final section, we analyze the ranks in LoRA based on the checkpoints from commonsense reasoning task. We analyzed the singular values of the increment $BA$ and give the following visualizations as shown in Fig. 6.

The LoRA updates exhibit a smooth, almost linear decay in log-scale, with no clear spectral gap. While for LAVA, the singular values are nearly flat up to an index of about 16–17, followed by a sharp drop of several orders of magnitude. In other words, LoRA tends to keep many useless and redundant directions despite the fact that the effective ranks are far fewer.

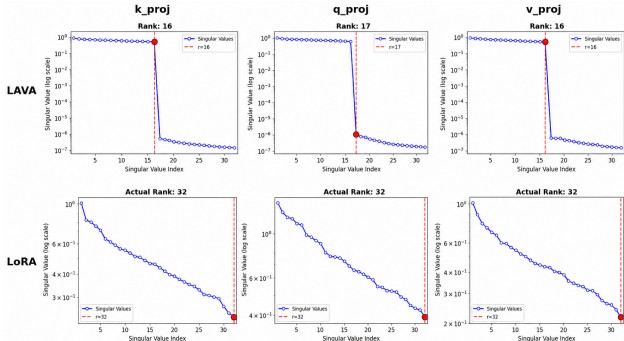

*Figure 6.* Singular values of LoRA and LAVA.

# 8. Conclusion

In this work, we first conducted a novel increment analysis experiment to empirically validate the existence of intrinsic dimensions in convolution layers, which is similar to that in LoRA. Inspired by the observations from the experiment, we proposed LAVA to generalize the update of the matrix and convolution together. Moreover, we analyzed the training dynamics and presented to use orthogonal regularization in this new parameterization form. We also provided the theoretical analysis that it can reduce the variance of trainable parameters. LAVA consistently outperforms LoRA for various downstreaming tasks and model architectures.

## Impact Statement

This paper presents work whose goal is to advance the field of Machine Learning. There are many potential societal consequences of our work, none which we feel must be specifically highlighted here.

## Acknowledgments

This work was supported by the National Natural Science Foundation of China (No. 62276182), and Tianjin Natural Science Foundation (Nos. 24JCYBJC01230, 24JCYBJC01460, 24JCZDJC01190).

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

# A. Appendix

## A.1. The Use of Large Language Models (LLMs)

LLMs are used to polish up the writing in the abstract and introduction parts.

## A.2. Proof of Theorem 1

**Theorem 1. Upper bound of Frobenius Norm $\|A(t)\|_F$ and $\|B(t)\|_F$.** If the optimizer is SGD, denote the full gradient as $\bar{G}(t)$, and the gradient calculated on a mini-batch as $G(t)$. Suppose $\mathbb{E}[G(t)] = \bar{G}(t)$, $\|\bar{G}(t)\|_2 \leq \mu$, and $\mathbb{E}[\|G(t) - \bar{G}(t)\|_2^2] \leq \sigma^2$. Denote $v(t) = \|A(t)\|_F^2 + \|B(t)\|_F^2$, $v_0 = \|A_0\|_F^2 + \|B_0\|_F^2$, and $\gamma = \frac{\mu^2 + \sigma^2}{2\lambda}$. Then, for any finite training horizon $T$, we have

$$\sup_{t \in [0,T]} \mathbb{E}[v(t)] \leq v_0 e^{\gamma T} + \frac{16\lambda^2 m}{\mu^2 + \sigma^2} \left(e^{\gamma T} - 1\right). \tag{8}$$

**Proof.**

Based on the orthogonal regularization on $A^\top A$, we define $D(t) = A(t)^\top A(t) - I$. The gradient flow can be written as

$$\begin{cases} A'(t) = -B(t)^\top G(t) - 4\lambda A(t)D(t), \\ B'(t) = -G(t)A(t)^\top. \end{cases} \tag{9}$$

Let $v(t) = \|A(t)\|_F^2 + \|B(t)\|_F^2$. Then

$$\begin{aligned} v'(t) &= 2\langle A, A' \rangle_F + 2\langle B, B' \rangle_F \\ &= -4\langle BA, G \rangle_F - 8\lambda \mathrm{tr}\left(A^\top A(A^\top A - I)\right). \end{aligned} \tag{10}$$

Using the exact identity

$$\begin{aligned} \mathrm{tr}\left(A^\top A(A^\top A - I)\right) &= \mathrm{tr}\left((A^\top A - I + I)(A^\top A - I)\right) \\ &= \|A^\top A - I\|_F^2 + \|A\|_F^2 - m, \end{aligned} \tag{11}$$

we have

$$v'(t) = -4\langle BA, G \rangle_F - 8\lambda \|A^\top A - I\|_F^2 - 8\lambda \|A\|_F^2 + 8\lambda m. \tag{12}$$

For the interaction term, by Cauchy's inequality and Young's inequality,

$$\begin{aligned} 4|\langle BA, G \rangle_F| &= 4|\langle A, B^\top G \rangle_F| \\ &\leq 4\|A\|_F \|B^\top G\|_F \\ &\leq 4\|A\|_F \|B\|_F \|G\|_2 \\ &\leq 8\lambda \|A\|_F^2 + \frac{\|G(t)\|_2^2}{2\lambda} \|B\|_F^2. \end{aligned} \tag{13}$$

Substituting the above inequality into $v'(t)$, and dropping the non-positive term $-8\lambda \|A^\top A - I\|_F^2$, we obtain

$$v'(t) \leq \frac{\|G(t)\|_2^2}{2\lambda} \|B(t)\|_F^2 + 8\lambda m \leq \frac{\|G(t)\|_2^2}{2\lambda} v(t) + 8\lambda m. \tag{14}$$

Based on the stochastic gradient assumptions, $\mathbb{E}[\|G(t)\|_2^2] \leq \mu^2 + \sigma^2$. Since $v(t)$ is determined by the current trajectory, we have

$$\frac{d}{dt}\mathbb{E}[v(t)] \leq \frac{\mu^2 + \sigma^2}{2\lambda} \mathbb{E}[v(t)] + 8\lambda m. \tag{15}$$

Denote $\gamma = \frac{\mu^2 + \sigma^2}{2\lambda}$. By Gronwall's inequality,

$$\begin{aligned} \mathbb{E}[v(t)] &\leq v_0 e^{\gamma t} + 8\lambda m \int_0^t e^{\gamma(t-s)} ds \\ &= v_0 e^{\gamma t} + \frac{8\lambda m}{\gamma} \left(e^{\gamma t} - 1\right) \\ &= v_0 e^{\gamma t} + \frac{16\lambda^2 m}{\mu^2 + \sigma^2} \left(e^{\gamma t} - 1\right). \end{aligned} \tag{16}$$

Taking the supremum over $t \in [0, T]$, we get

$$\sup_{t \in [0,T]} \mathbb{E}[v(t)] \leq v_0 e^{\gamma T} + \frac{16\lambda^2 m}{\mu^2 + \sigma^2} \left(e^{\gamma T} - 1\right). \tag{17}$$

## A.3. Proof of Theorem 2

**Theorem 2. Convergence rate under orthogonal regularization**. If the optimizer is SGD, when the regularization strength term is set to $\lambda$, $A$ will get converged to orthogonal matrix $M$ at a speed of $\mathcal{O}(\frac{\mu+\sigma}{\lambda})$.

**Proof.**

If the optimizer is SGD, denote $D_k = A_k^\top A_k - I$, $G_k = \nabla_W L(A_k, B_k)$, and $C_k = A_k^\top A_k$. Then the parameter update has the following form:

$$\begin{cases} A_{k+1} \leftarrow A_k - \eta[B_k^\top G_k + 4\lambda A_k(A_k^\top A_k - I)] = A_k(I - 4\eta\lambda D_k) - \eta B_k^\top G_k, \\ B_{k+1} \leftarrow B_k - \eta G_k A_k^\top. \end{cases} \tag{18}$$

Then

$$\begin{aligned} C_{k+1} &= A_{k+1}^\top A_{k+1} \\ &= [(I - 4\eta\lambda D_k)^\top A_k^\top - \eta G_k^\top B_k][A_k(I - 4\eta\lambda D_k) - \eta B_k^\top G_k] \\ &= (I - 4\eta\lambda D_k)^\top C_k(I - 4\eta\lambda D_k) \\ &\quad - \eta(I - 4\eta\lambda D_k)^\top A_k^\top B_k^\top G_k - \eta G_k^\top B_k A_k(I - 4\eta\lambda D_k) + \mathcal{O}(\eta^2 I) \\ &= C_k - 4\eta\lambda D_k C_k - 4\eta\lambda C_k D_k - \eta(A_k^\top B_k^\top G_k + G_k^\top B_k A_k) + \mathcal{O}(\eta^2 I). \end{aligned} \tag{19}$$

When training LoRA, the learning rate $\eta$ is often set to be small, thus the discrete optimization process can be interpreted as a numerical integration of a continuous-time dynamical system, or **gradient flow**. Denote $C(t) = A(t)^\top A(t)$, $D(t) = C(t) - I = A^\top A - I$, and $H = A^\top B^\top G + G^\top BA$. Since $C = D + I$, $C$ and $D$ commute. Then we get the following formulas:

$$\begin{cases} \frac{dA}{dt} = -B^\top G - 4\lambda AD, \\ \frac{dB}{dt} = -GA^\top, \\ \frac{dC}{dt} = -4\lambda DC - 4\lambda CD - H, \\ \frac{dD}{dt} = -8\lambda(D^2 + D) - H. \end{cases} \tag{20}$$

Let $S(t) = \|A^\top A - I\|_F^2 = \|D\|_F^2 = \text{tr}(D^2)$ and $d(t) = \|D(t)\|_F$. Then

$$\begin{aligned} \frac{dS(t)}{dt} &= 2\langle D, D'\rangle_F \\ &= -16\lambda\text{tr}(D(D^2 + D)) - 2\langle D, H\rangle_F. \end{aligned} \tag{21}$$

For the perturbation term,

$$\begin{aligned} |\langle D, H\rangle_F| &\leq \|D\|_F\|H\|_F \\ &\leq \|D\|_F\left(\|A^\top B^\top G\|_F + \|G^\top BA\|_F\right) \\ &\leq 2\|D\|_F\|A\|_F\|B\|_F\|G\|_2. \end{aligned} \tag{22}$$

Since

$$\text{tr}(D(D^2 + D)) = \text{tr}(D^3) + \text{tr}(D^2) \geq \|D\|_F^2 - \|D\|_F^3, \tag{23}$$

we obtain

$$S'(t) \leq 4d(t)\|A\|_F\|B\|_F\|G\|_2 - 16\lambda(d^2(t) - d^3(t)). \tag{24}$$

Since $S'(t) = 2d(t)d'(t)$ when $d(t) > 0$, we have

$$d'(t) \leq 2\|A\|_F\|B\|_F\|G\|_2 - 8\lambda d(t) + 8\lambda d^2(t). \tag{25}$$

For any finite training horizon $[0, T]$, denote

$$K_T = \sup_{t\in[0,T]} 2\|A(t)\|_F\|B(t)\|_F\|G(t)\|_2. \tag{26}$$

Then

$$d'(t) \leq 8\lambda d^2(t) - 8\lambda d(t) + K_T. \tag{27}$$

We consider the following Riccati comparison equation:

$$y'(t) = 8\lambda y^2(t) - 8\lambda y(t) + K_T. \tag{28}$$

The corresponding characteristic equation is

$$8\lambda y^2 - 8\lambda y + K_T = 0. \tag{29}$$

When $K_T < 2\lambda$, it has two real roots:

$$r_1 = \frac{1}{2}\left(1 - \sqrt{1 - \frac{K_T}{2\lambda}}\right), \qquad r_2 = \frac{1}{2}\left(1 + \sqrt{1 - \frac{K_T}{2\lambda}}\right). \tag{30}$$

Thus, the comparison equation can be written as

$$y'(t) = 8\lambda(y(t) - r_1)(y(t) - r_2). \tag{31}$$

When $0 \le y(t) < r_1$, we have $y'(t) > 0$; when $r_1 < y(t) < r_2$, we have $y'(t) < 0$. Therefore, $r_1$ is a stable equilibrium of the Riccati comparison system. By the Comparison Principle, if $d(0) < r_2$, then the true deviation trajectory is controlled by this auxiliary trajectory. Thus

$$d(t) \le \max\{d(0), r_1\}, \qquad t \in [0, T]. \tag{32}$$

Moreover, the stable radius is

$$r_1 = \frac{1}{2}\left(1 - \sqrt{1 - \frac{K_T}{2\lambda}}\right). \tag{33}$$

When $\frac{K_T}{2\lambda} \ll 1$, based on Taylor expansion,

$$\sqrt{1 - \frac{K_T}{2\lambda}} = 1 - \frac{K_T}{4\lambda} + \mathcal{O}\left(\frac{K_T^2}{\lambda^2}\right). \tag{34}$$

Therefore,

$$r_1 = \frac{K_T}{8\lambda} + \mathcal{O}\left(\frac{K_T^2}{\lambda^2}\right). \tag{35}$$

Denote

$$M_T = \left(\sup_{t \in [0,T]} \mathbb{E}[v^2(t)]\right)^{1/2}, \tag{36}$$

since

$$2\|A\|_F\|B\|_F\|G\|_2 \le (\|A\|_F^2 + \|B\|_F^2)\|G\|_2 = v(t)\|G(t)\|_2, \tag{37}$$

by Cauchy's inequality and the stochastic gradient assumption,

$$\sup_{t \in [0,T]} \mathbb{E}\left[2\|A(t)\|_F\|B(t)\|_F\|G(t)\|_2\right] \le M_T \left(\sup_{t \in [0,T]} \mathbb{E}[\|G(t)\|_2^2]\right)^{1/2}$$

$$\le M_T\sqrt{\mu^2 + \sigma^2} \le M_T(\mu + \sigma). \tag{38}$$

Combining this with the Riccati stable radius $r_1 = \mathcal{O}(K_T/\lambda)$, we obtain the expected finite-horizon scaling

$$\|A^\top A - I\|_F = \mathcal{O}_T\left(\frac{\mu + \sigma}{\lambda}\right), \tag{39}$$

where the hidden constant depends on $T$ and the moment bound of $v(t)$.

## A.4. Proof of Theorem 3: Upper bound of gradient variance

**Proof.**

Let $G = \nabla_W L \in \mathbb{R}^{m \times n}$, $\Sigma_G = \mathbb{E}[(G - \mathbb{E}G)^\top(G - \mathbb{E}G)] \in \mathbb{R}^{n \times n}$. Based on the definition of variance:

$$\begin{aligned}
\mathrm{Var}(\nabla_B L) &= \mathbb{E}\|GA^\top - \mathbb{E}[GA^\top]\|_F^2 \\
&= \mathbb{E}[tr((GA^\top - \mathbb{E}[GA^\top])^\top(GA^\top - \mathbb{E}[GA^\top]))] \\
&= tr(\mathbb{E}[(GA^\top - \mathbb{E}[GA^\top])^\top(GA^\top - \mathbb{E}[GA^\top])]) \\
&= tr(A\mathbb{E}[(G - \mathbb{E}G)^\top(G - \mathbb{E}G)]A^\top) \\
&= tr(A\Sigma_G A^\top) \\
&= tr(A^\top A\Sigma_G) \le tr(\|A^\top A\|_2\Sigma_G) \quad (*) \\
&\le (1 + E_A)tr(\Sigma_G).
\end{aligned} \tag{40}$$

Next we prove $(*)$: if $B \succeq 0$, then $tr(AB) \le \|A\|_2 tr(B)G_{max}$.

Since $B$ is positive semi-definite, $B$ can be diagonalized into the following form: $B = Q\Lambda Q^\top$. Denote matrix $D = Q^\top AQ$, then,

$$
\begin{aligned}
tr(AB) = tr(AQ\Lambda Q^\top) &= tr(Q^\top AQ\Lambda) \\
&= tr(D\Lambda) = \sum_i d_{ii} \cdot \lambda_i \\
&\leq \sum_i \|D\|_2 \lambda_i \\
&= \sum_i \|A\|_2 \cdot \lambda_i \\
&= \|A\|_2 tr(B).
\end{aligned}
\tag{41}
$$

## A.5. Toy experiment

We finetune ResNet18, ResNet34, VGG11, AlexNet, GoogLeNet and ConvNeXt on CIFAR10 using the optimizer AdamW (Loshchilov & Hutter, 2019). The learning rate is set to be 1e-3, betas for the optimizer are (0.9, 0.999), and weight decay is set to be 0. In this experiment, to analyze the low-rank subspaces, we don't set the orthogonal regularization term. Table 10 lists the results of three methods in all the models.

*Table 10.* Comparisons of reshape-involved, reshape-free and full-finetuning methods on CIFAR10.

| Model | Metric | AlexNet | VGG11 | ResNet34 | GoogLeNet | ResNet18 | ConvNeXt |
|---|---|---|---|---|---|---|---|
| Reshape-involved | Trainable parameters(%) | 0.13 | 0.09 | 1.05 | 2.32 | 1.01 | 3.12 |
| | Accuracy | 74.21 | 60.47 | 78.58 | 79.44 | 76.16 | 76.09 |
| Reshape-free | Trainable parameters(%) | 0.08 | 0.04 | 0.25 | 1.66 | 0.28 | 0.15 |
| | Accuracy | 87.27 | 90.09 | 93.85 | 93.52 | 92.37 | 98 |
| FT | Trainable parameters(%) | - | - | - | - | - | - |
| | Accuracy | 86.81 | 90.94 | 92.63 | 93.83 | 92.88 | 95.34 |

In Table. 11, we test more valid unfolding choices.

*Table 11.* Comparisons between additional reshape-involved and reshape-free methods on CIFAR10.

| Model | $out \times (in \times h \times w)$ | $h \times (out \times in \times w)$ | $w \times (out \times in \times h)$ | Reshape-Free |
|---|---|---|---|---|
| ResNet18 | 76.16 | 77.69 | 80.56 | 92.37 |
| ResNet34 | 74.87 | 80.40 | 82.19 | 93.85 |
| VGG11 | 64.36 | 81.31 | 82.50 | 90.09 |
| ConvNeXt | 76.09 | 88.01 | 88.32 | 98 |
| GoogLeNet | 79.44 | 83.47 | 81.64 | 93.52 |

## A.6. NLU Experiment Details

The GLUE benchmark (General Language Understanding Evaluation) is a widely used collection of datasets and evaluation metrics designed to assess the performance of natural language understanding (NLU) models. Following (Wu et al., 2024a), we split the publicly available validation dataset into two parts: if the validation dataset is larger than 2K, then 1K is chosen as a new validation set; otherwise, half of it is chosen as new validation set. Table 12 shows the detailed split and metric used for each dataset in GLUE benchmark. Here, MCC represents the Matthews correlation coefficient, ACC represents accuracy, and CORR represents Pearson correlation coefficient.

We provide details of the chosen baselines below:

*Full Finetune (FT)*: Full fine-tuning is a common approach for adaptation. During adaptation, all parameters of the model undergo gradient updates.

*Table 12.* Splits and metrics of the GLUE benchmark.

| Split Sizes | MNLI | SST-2 | MRPC | CoLA | QNLI | QQP | RTE | STS-B |
|---|---|---|---|---|---|---|---|---|
| **#Train** | 393K | 67K | 3.7K | 8.5K | 105K | 364K | 2.5K | 5.7K |
| **#Validation** | 1K | 436 | 204 | 522 | 1K | 1K | 139 | 750 |
| **#Test** | 8K | 436 | 204 | 521 | 4.5K | 39K | 138 | 750 |
| **Metric** | ACC | ACC | ACC | MCC | ACC | ACC | ACC | CORR |

*Adapter*: Adapter (Houlsby et al., 2019) inserts lightweight and trainable modules between two sub-layers of the transformer.

*Adapter-FFN*: Adapter-FFN (Pfeiffer et al., 2021) is a variant of Adapter method, it inserts lightweight and trainable modules after FFN sub-layer in transformer architecture.

*BitFit*: Bias-only (BitFit) is an early work proposed by (Ben Zaken et al., 2022): it only tuns bias-terms of the model.

*LoReFT*: Low-rank Linear Subspace ReFT, or LoReFT (Wu et al., 2024b), learns task-specific interventions on hidden representations to adapt the pre-trained model on down-streaming tasks.

*RED*: Representation Editing or (RED) (Wu et al., 2024a), similar to LoReFT, is a method to modify representations generated at some layers using scaling and biasing operations.

*LoRA*: Vanilla LoRA, trains two matrices, one for down-projection the input into low-rank spaces, and one for up-projecting it back.

*DeLoRA*: (Bini et al., 2025) decouples the weight into two components, direction and normalization. By bounding the distance of the transformation, it can enhance the robustness.

For all the experiments, the maximum sequence length is set to be 512. Other hyperparameters are chosen by grid search. All the experiment results are averaged on random seeds 42, 43, 44 45 and 46. Hyperparameters are reported in Table 13.

*Table 13.* Hyperparameters for LAVA in NLU

| Hyperparameters | MNLI | SST-2 | MRPC | CoLA | QNLI | QQP | RTE | STS-B |
|---|---|---|---|---|---|---|---|---|
| Rank | | | | 8 | | | | |
| Dropout | | | | 0.1 | | | | |
| alpha | | | | 8 | | | | |
| Max Seq | | | | 512 | | | | |
| Learning Rate $\eta$ | 5e-4 | 5e-4 | 3e-4 | 4e-4 | 4e-4 | 5e-4 | 5e-4 | 4e-4 |
| Epoch | 30 | 20 | 20 | 80 | 25 | 25 | 80 | 40 |
| Batch size | 16 | 8 | 4 | 32 | 32 | 64 | 32 | 32 |
| orthogonal regularization | 0.3 | 1.0 | 1.0 | 0.05 | 0.025 | 0.1 | 0.1 | 0.05 |

### A.7. Commonsense Reasoning Experiment

For fair comparison, we set the rank to be 32, which is the same as experiments in (Hu et al., 2023), and we only tune the learning rate and orthogonal regularization term $\lambda$.

Table 14 shows all the results on eight datasets for our method. LAVA almost outperforms the performances of all the baselines on every dataset.

### A.8. Semantic Segmentation Experiment Details

#### A.8.1. HYPERPARAMETERS FOR THE EXPERIMENT

In semantic segmentation experiments, we follow the setting of Conv-LoRA (Zhong et al., 2024) and test our method on datasets about polyp, skin lesion, camouflaged object, leaf disease and shadow segmentation. Table 15 lists out the mapping

*Table 14.* Performance comparisons on Gemma2-2b, LLaMA3-1b and LLaMA2-7b on eight commonsense reasoning datasets. Best result is marked **bold**.

| Model | Method | # Params (%) | BoolQ | PIQA | SIQA | HellaSwag | WinoGrande | ARC-e | ARC-c | OBQA | **Avg.** |
|---|---|---|---|---|---|---|---|---|---|---|---|
| ChatGPT | - | - | 73.1 | 85.4 | 68.5 | 78.5 | 66.1 | 89.8 | 79.9 | 74.8 | 77.0 |
| Gemma2-2b | LoRA | 1.07 | 68.5 | 80.5 | 77.2 | 86.9 | 78.5 | 81.8 | 66.0 | **79.6** | 77.4 |
| | VeRA | 0.02 | 65.6 | 75.4 | 74.6 | 59.3 | 72.8 | 80.0 | 63.0 | 70.6 | 70.1 |
| | LoRA+ | 1.07 | **68.6** | **81.1** | **77.7** | 89.4 | 77.6 | **84.1** | 66.8 | 75.0 | 77.5 |
| | DoRA | 1.09 | 67.5 | 80.7 | **77.7** | 87.2 | 79.2 | 81.6 | 66.7 | 77.8 | 77.3 |
| | LAVA | 1.07 | 68.5 | 80.8 | 77.6 | 88.8 | **80.4** | 83.2 | 66.6 | 78.6 | **78.2** |
| LLaMA3-3b | LoRA | 1.02 | 64.3 | **85.3** | 74.4 | **92.4** | 78.4 | 76.8 | 66.9 | 76.0 | 76.8 |
| | DoRA | 1.03 | 68.2 | 84.7 | 78.3 | 90.7 | **79.6** | **84.2** | **71.9** | **78.8** | 79.5 |
| | LAVA | 1.02 | **71.4** | 84.9 | **79.4** | 91.2 | 79.2 | 84.0 | 69.7 | **78.8** | **79.8** |
| LLaMA2-7b | LoRA | 0.83 | 69.8 | 79.9 | **79.5** | 83.6 | **82.6** | 79.8 | 64.7 | **81.0** | 77.6 |
| | LAVA | 0.83 | **71.9** | **84.2** | 78.3 | **86.6** | 81.9 | **83.4** | **68.6** | 80.8 | **79.5** |

between training and testing datasets. For detailed descriptions and dataset configurations, please refer to (Zhong et al., 2024).

In this experiment, we compare LAVA against two baselines: LoRA and Conv-LoRA. Conv-LoRA introduces MoE structure (Shazeer et al., 2017) into LoRA. On each path of MoE, Conv-LoRA injects lightweight and trainable convolution layers to extract features at different scale. Such a method could introduce image-related inductive bias at different scales, which benefits the performances.

*Table 15.* Details of train and testing datasets

| **Train** | Polyp | ISIC 2017 | CAMO | SBU | Leaf |
|---|---|---|---|---|---|
| **Test** | CVC-612 | ISIC 2017 | CAMO | SBU | Leaf |

Tables 16, 17 and 18 provide hyperparameters used in this experiment.

*Table 16.* Hyperparameters for Conv-LoRA used in semantic segmentation

| Dataset | Rank | # experts | Batch size | Learning Rate $\eta$ | Epoch | Metric | Loss |
|---|---|---|---|---|---|---|---|
| Polyp | 3 | 8 | 4 | 1e-4 | 30 | sm | structure_loss |
| CAMO | 3 | 8 | 4 | 1e-4 | 20 | sm | structure_loss |
| Leaf | 3 | 8 | 4 | 3e-4 | 30 | iou | structure_loss |
| ISIC2017 | 3 | 8 | 4 | 1e-3 | 30 | iou | structure_loss |
| SBU | 3 | 8 | 4 | 1e-4 | 10 | ber | balanced_bce |

*Table 17.* Hyperparameters for LoRA used in semantic segmentation

| Dataset | Rank | Alpha | Batch size | Learning Rate $\eta$ | Epoch | Metric | Loss |
|---|---|---|---|---|---|---|---|
| Polyp | 3 | 32 | 4 | 1e-4 | 30 | sm | structure_loss |
| CAMO | 3 | 32 | 4 | 1e-4 | 20 | sm | structure_loss |
| Leaf | 3 | 32 | 4 | 3e-4 | 30 | iou | structure_loss |
| ISIC2017 | 32 | 3 | 4 | 1e-3 | 30 | iou | structure_loss |
| SBU | 3 | 32 | 4 | 1e-4 | 10 | ber | balanced_bce |

*Table 18.* Hyperparameters for LAVA used in semantic segmentation

| Dataset | Rank | Alpha | Orthogonal strength Multiplier $\lambda$ | Batch size | Learning Rate $\eta$ | Epoch | Metric | Loss |
|---|---|---|---|---|---|---|---|---|
| Polyp | 3 | 32 | 0.5 | 4 | 1e-4 | 30 | sm | structure_loss |
| CAMO | 3 | 32 | 0.1 | 4 | 1e-4 | 20 | sm | structure_loss |
| Leaf | 3 | 32 | 0.1 | 4 | 3e-4 | 30 | iou | structure_loss |
| ISIC2017 | 3 | 32 | 1.0 | 4 | 1e-3 | 30 | iou | structure_loss |
| SBU | 3 | 32 | 0.1 | 4 | 1e-4 | 10 | ber | balanced_bce |

### A.8.2. QUANTITATIVE COMPARISONS IN SEMANTIC SEGMENTATION EXPERIMENT

*Table 19.* Performance comparison across different domains. Best result is marked **bold**. The second highest value is marked using underline. All the experiment results are averaged on random seeds 42, 43 and 44.

| Method | Ratio | Medical | | | | Natural Images | | | | Agriculture | |
|---|---|---|---|---|---|---|---|---|---|---|---|
| | | CVC-612 | | ISIC 2017 | | CAMO | | | SBU | Leaf | |
| | | $S_\alpha \uparrow$ | $E_\phi \uparrow$ | Jac $\uparrow$ | Dice $\uparrow$ | $S_\alpha \uparrow$ | $E_\phi \uparrow$ | $F_\beta \uparrow$ | BER $\downarrow$ | IoU $\uparrow$ | Dice $\uparrow$ |
| LoRA | 4.00 / 0.62% | $90.3_{\pm 0.63}$ | $91.8_{\pm 0.92}$ | $\underline{77.3}_{\pm 0.66}$ | $\underline{87.2}_{\pm 0.42}$ | $\mathbf{88.6}_{\pm 0.52}$ | $\underline{92.4}_{\pm 0.65}$ | $83.5_{\pm 1.13}$ | $2.86_{\pm 0.11}$ | $72.2_{\pm 0.84}$ | $83.9_{\pm 0.56}$ |
| Conv-LoRA | 4.02 / 0.63% | $\underline{90.4}_{\pm 0.62}$ | $92.1_{\pm 0.29}$ | $77.2_{\pm 0.05}$ | $87.1_{\pm 0.03}$ | $88.2_{\pm 0.25}$ | $91.8_{\pm 0.16}$ | $\underline{83.6}_{\pm 0.06}$ | $\underline{2.78}_{\pm 0.03}$ | $\underline{73.1}_{\pm 0.21}$ | $\underline{84.5}_{\pm 0.14}$ |
| LAVA | 4.00 / 0.62% | $\mathbf{91.0}_{\pm 0.73}$ | $\mathbf{93.4}_{\pm 0.63}$ | $\mathbf{77.9}_{\pm 0.41}$ | $\mathbf{87.6}_{\pm 0.26}$ | $\mathbf{88.6}_{\pm 0.08}$ | $\mathbf{92.6}_{\pm 0.42}$ | $\mathbf{83.7}_{\pm 0.42}$ | $\mathbf{2.64}_{\pm 0.11}$ | $\mathbf{73.6}_{\pm 0.25}$ | $\mathbf{84.8}_{\pm 0.17}$ |

Table 19 provides the detailed quantitative results for all three methods. LAVA could improve the segmentation results further while keeping the trainable parameters at the same level compared with other two PEFT methods.

## A.9. Depth Estimation Experiment Details

In this experiment, we finetune Depth-Anything in metric depth estimation. Metric depth estimation is a field that predicts the absolute distance in real-world units from the camera to each pixel in an image. Firstly, we provide some descriptions of the chosen baselines (unmentioned baselines have been discussed in the main text):

- *SVDiff*: SVDiff can be recognized as a general framework to finetune both convolution and matrix. For convolution layers, it flattens the tensor $\mathcal{X} \in \mathbb{R}^{c_{out} \times c_{in} \times h \times w}$ into its matrix form $X \in \mathbb{R}^{c_{out} \times (c_{in} \times h \times w)}$, and then perform SVD and only tunes its singular values.

- *VeRA*: VeRA (Kopiczko et al., 2024) replaces learnable low-rank matrices (i.e., $A$ and $B$ in LoRA) with fixed random matrices and trains only two small scaling vectors $\lambda_d$ (per-rank) and $\lambda_b$ (per-output-dimension).

### A.9.1. HYPERPARAMETERS FOR DEPTH-ESTIMATION EXPERIMENT

To finetune convolution layers, LoRA reshapes the convolution weights into its matrix form and then tunes the matrix accordingly. For LoRA, we conduct two experiments: one to only tune the encoder part, and the other to tune both the encoder and decoder of the model. For SVDiff, we tune both components. We believe that this setting could prove the necessity to tune convolution parts in pixel-level granularity tasks. The batch size is set to be 4, and we run each baseline method for 10 epochs. For SVDiff, the learning rate is set to be 0.000161 at the beginning, and weight decay is set to be 0.01. For LoRA (both trained on encoder and encoder+decoder) and LAVA, the learning rate is set be 0.000161, and weight decay is the same with the setting in SVDiff. Additionally, the orthogonal regularization term of LAVA is set to be 0.01.

### A.9.2. MORE RESULTS TO COMPARE LAVA AGAINST FULL-FINETUNING

We provide more qualitative results in Fig. 7 to showcase LAVA's superiority in difficult situations compared with full fine-tuning. As discussed in the Qualitative Comparison parts before, LAVA could successfully recognize and calculate the real distance in regions with strong ambient light, perspective in transparent objects, and reflections. For example, in the first column of Fig. 7, the induction cooker on the table reflects sunlight. In this case, LAVA could still predict the item on the table, while full-finetuning fails to do so.

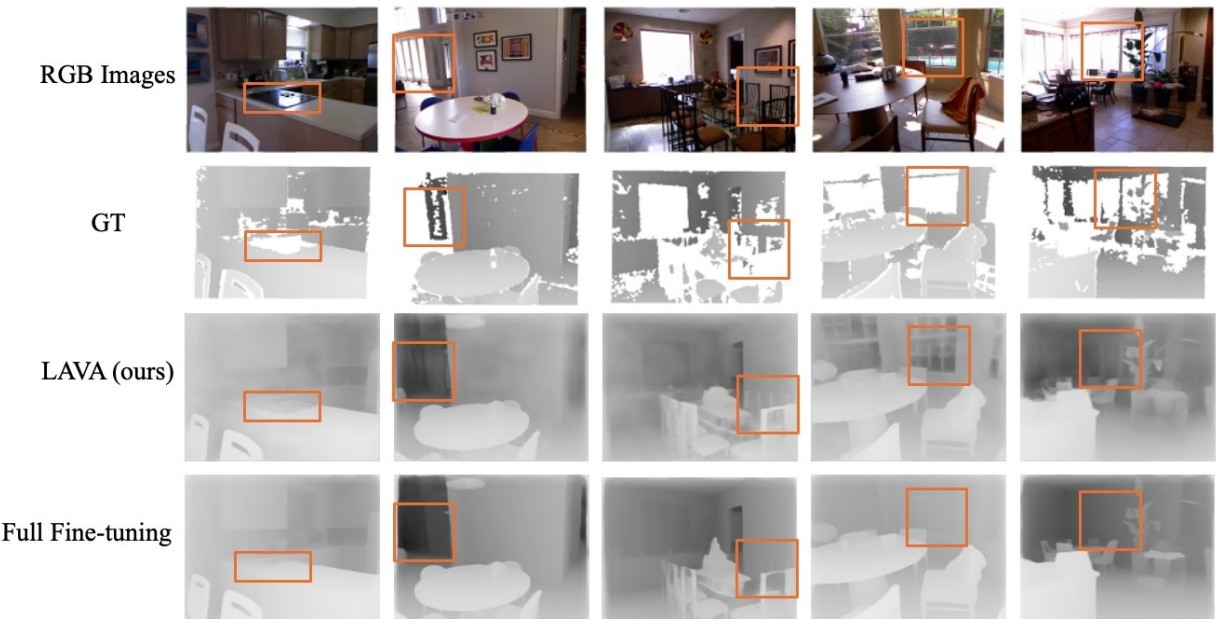

*Figure 7.* More qualitative results between LAVA and full-finetuning.

### A.9.3. MORE RESULTS TO COMPARE PEFT METHODS IN THE DEPTH-ESTIMATION TASK

In Fig. 8, it lists out the generated images from all the methods. Compared DoRA with LoRA, DoRA shows that weight normalization can help learn features better. However, the generated image is still relatively blurry, meaning that DoRA is not sufficient in processing reshape-involved tuning. Conv-Adapter is another competing baseline, but as shown in the fourth column, the outlines of the object have unwelcoming artifacts.

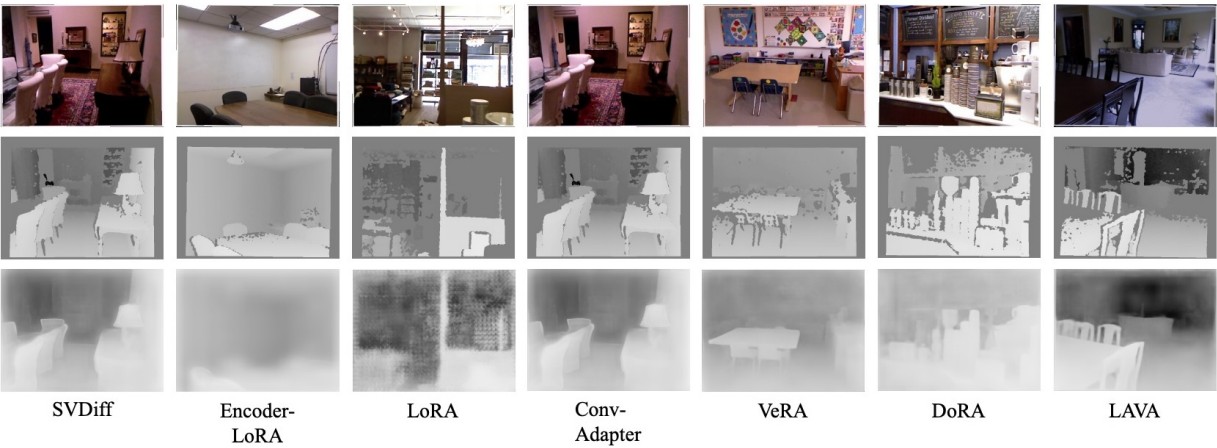

*Figure 8.* More qualitative results between PEFT methods.

### A.10. Text-To-Image Generation Experiment Details

FID (Heusel et al., 2017) and CLIP score are two widely used evaluation metrics for image generation models. Compared with metrics to compare pixels directly, they compare feature representations extracted from pre-trained Inception models. In Table 20, we provide FID and CLIP scores for LoRA and LAVA in five different random seed settings.

*Table 20.* FID comparisons

| Metric | LoRA | LAVA |
|--------|------|------|
| **FID Score**↓ | 480.41 | 429.38 |
| **CLIP Score**↑ | 0.57 | 0.60 |

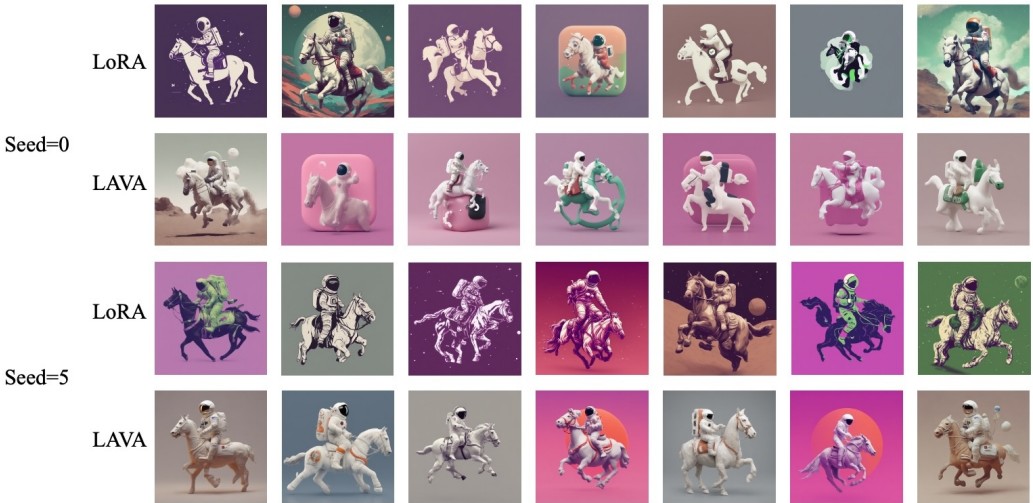

*Figure 9.* Comparisons of generated images from LoRA and LAVA.

## A.11. Robustness of hyper-parameters

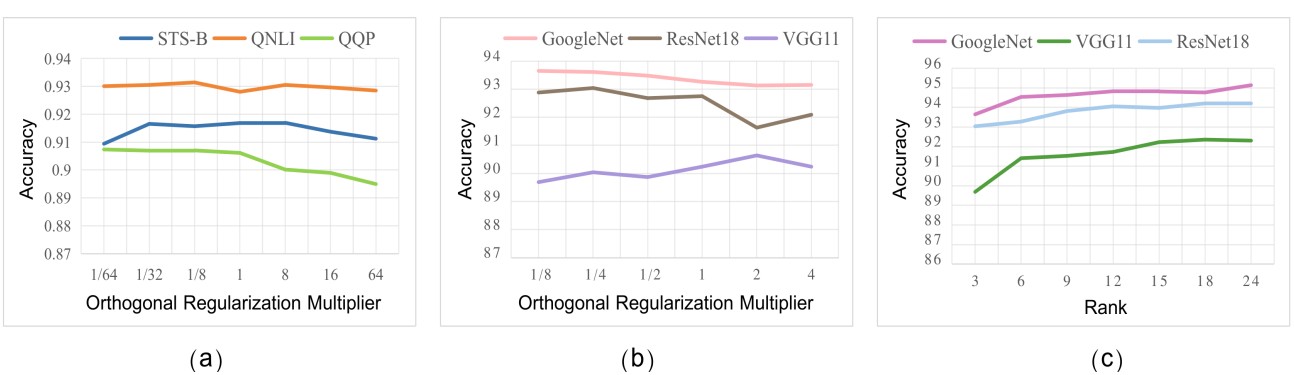

*Figure 10.* Hyperparameter analysis for LAVA. (a): Performances on NLU with different $\lambda$ settings. (b): Classification performances on CIFAR10 with different $\lambda$ settings on three model architectures. (c): Classification performances on CIFAR10 with different rank $R$ settings on three model architectures.

## A.12. Effects of different regularization methods on pre-trained convolutional networks.

For fair comparison, we fix the orthogonal regularization strength $\lambda$ at 0.025, and apply it on different blocks to compare the performances. As is shown in Fig. 11, we empirically find that imposing orthogonality on the channel-mode factors U and V (corresponding to output and input channels) yields the largest performance gains, while orthogonalizing the spatial factors X and Y brings marginal or even negative effects.

## A.13. Variance analysis

In this section, we provide additional variance analysis of Roberta-base on the MRPC and STSB datasets from the GLUE benchmark.

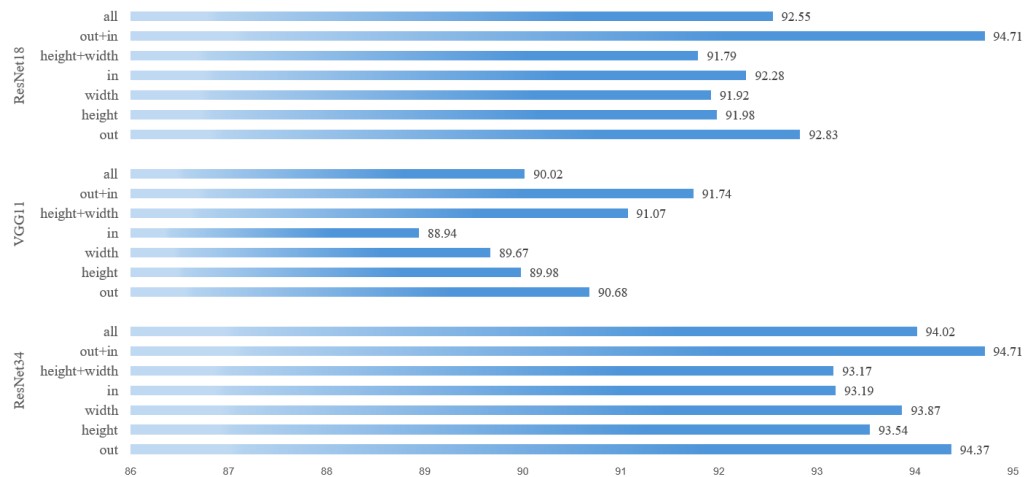

*Figure 11.* Comparisons between different orthogonal regularization methods.

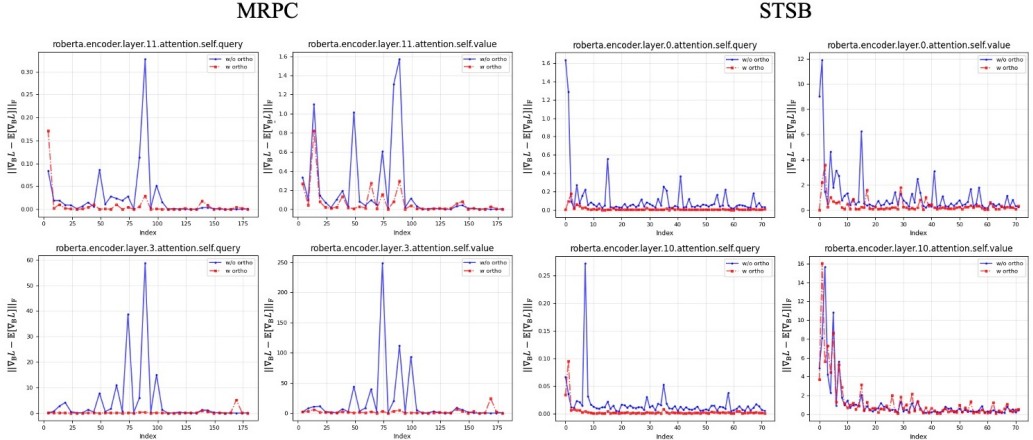

*Figure 12.* Variance analysis of the gradient.

