# OpenReview forum: "LAVA: A Unified Framework for Finetuning Language and Vision Models"
_ICML.cc/2026/Conference — ICML 2026 regular_

### Official Review · Reviewer_BWFH · 2026-03-10

**Soundness:** 3
**Presentation:** 2
**Significance:** 2
**Originality:** 2
**Overall Recommendation:** 4
**Confidence:** 3

**Summary:**

In this paper, the authors propose a new PEFT method (which they call LAVA) based on two main ideas: reshape-free tensor deomposition for convolutional layers (which reduces to standard LoRA for matrices) and column-orthogonal regularization on factor matrices which stabilize the training. Furthermore, the later also has some theoretical analysis. The authors combine these two ideas and compare LAVA with several other PEFT methods on different tasks, usually outperforming the other methods.

**Compliance With Llm Reviewing Policy:**

Affirmed.

**Final Justification:**

I thank the authors for providing a strong rebuttal.

My concerns were on the following:

1) Lack of a proper ablation study - which the authors provided (albeit to another reviewer but noted here).

2) Comparison on other benchmarks, where LaVA performs roughly on par with DoRA.

3) Better theoretical justification - provided by the authors in the rebuttal.

Overall, my main concerns have been addressed and I am increasing my score to Weak Accept.

**Key Questions For Authors:**

My main point is the ablation that I put in (1) under the Limitations section.

**Limitations:**

I think the paper has the following points that might need some improvement:

1) The two main contributions of the paper are not properly ablated (this also corresponds to Key Question 1), thus we do not know ehre do the gains come from. I think that there should be an ablation as the following:

Setting                                             Attention                      Convolution

LoRA                                                LoRA                           Reshape-involved LoRA

LoRA + orthogonal regularization    LoRA + orthogonal      Reshape-involved LoRA + orthogonal

LAVA no orthogonal regularization  LoRA                            Reshape-free decomposition

Full LAVA                                         LoRA + orthogonal       Reshape-free + orthogonal

Please correct me if I have not done a proper matching.

 2) The gains in the NLP task are somehow marginal and potentially incomplete.  GLUE: +0.3% over DeLoRA (within noise). Commonsense: LLaMA2-7b compared only against vanilla LoRA — no DoRA, LoRA+, or DeLoRA at 7B parameter. Since LAVA = LoRA + ortho reg for matrices, these results only evaluate the regularizer, and the evidence it helps substantially is weak. Furthermore, LLaMA-2 is quite outdated, thus it is quite hard to really get the value of these results. Probably would have been much better to compare on some better benchmark (e.g., latest versions of QWen or InternLM).

3) While I appreciate the theory and to the best of my understanding is correct, I also think it is quite loose. Theorem 1's norm bound depends on worst-case gradient norm. Theorem 2 drops the quadratic term in the proof. Thus, I have a hard understanding on the practical utility of it.

4) (More minor) It seems to me that the hyperparameter for orthogonal regularization must be perfectly tuned for the method to work. We can see that depending on the task it has been really tuned and for some task (SST-2, MRPC) where is set to 1, is 40 times higher than say in QNLI where it is set to 0.025. I wouldn't penalize too much about this though

**Strengths And Weaknesses:**

I think the paper has the following strengths:

1) Experimental section is pretty good.

- The reshape-free argument is quite convincing. In Section 3, the authors show a significant accuracy gain while using half the trainable parameters across six different architectures. While some architectures are extremely outdated, they also compared in decent CNN architectures such as ConvNext.

- Similarily, the depth estimation results are also quite good. LAVA outperforms LoRA and DoRA in all tasks there while using fewer parameters. It also reaches better results than VeRA albeit it uses more parameters.

- Furthermore, on both common reasoning and Roberta benchmarks, LAVA gets the best overall results (when you take into consideration both the number of parameters and average accuracy).

2) There is some interesting theoretical justifications such as the gradient flow framework devised in Theorems 1-3 that connect orthogonal regularization to parameter norms and the later singulal value analysis. In both cases these are principled justifications rather than just empirical ones.

---

> ### Author Rebuttal · Authors · 2026-03-30
>
> We greatly appreciate the reviewer’s careful reading and constructive feedback. Below we try to address the questions and clarify the relevant points.
>
> # 1. Lack of ablation study
>
> Due to context limit, we encourage reviewer to refer to our responses to Reviewer R6yf there (Sec: Clean ablation study).
>
> ---
>
> # 2. Comparisons on better benchmarks
>
> We provided comparisons between LAVA and other baselines on LLaMA3.2-3B in the following table:
>
> | Method | boolq | ARC-c | Arc-e | hellaswag | piqa  | siqa  | winogrande | openbookqa | avg   |
> | ------ | ----- | ----- | ----- | --------- | ----- | ----- | ---------- | ---------- | ----- |
> | LoRA   | 64.31 | 66.89 | 76.81 | 92.39     | 85.26 | 74.36 | 78.37      | 76.00      | 76.80 |
> | DoRA   | 68.23 | 71.93 | 84.22 | 90.71     | 84.66 | 78.25 | 79.56      | 78.80      | 79.54 |
> | LAVA   | 71.38 | 69.71 | 84.01 | 91.15     | 84.87 | 79.43 | 79.16      | 78.80      | 79.81 |
>
> ---
>
> # 3. Utility of the theory presented in the manuscript
>
> We deeply appreciate the reviewer's evaluation of our theoretical appendix. We agree that our theoretical presentation can be significantly strengthened. While our original proofs successfully capture the correct asymptotic convergence behaviors and intuitions of the LAVA framework, we acknowledge that the bounding techniques employed were conservative and simplified for brevity.
>
> ## a. Refining Theorem 1: Tightening the bound
>
> We agree that $G_{max}$ is a loose global bound. To incorporate the gradient variance assumptions $(\mu, \sigma)$ directly, we strictly analyze the system's energy functional $v(t) = \Vert A(t)\Vert_F^2 + \Vert B(t)\Vert _F^2$:
> $$
> v'(t) = -4\langle AB, G \rangle - 8\lambda \text{Tr}\big(A^\top A(A^\top A - I)\big).
> $$
> In which $\text{Tr}\big(A^\top A(A^\top A - I)\big) = \Vert A^\top A - I\Vert _F^2 + \Vert A\Vert _F^2 - r$.
> By applying Young's inequality, we can bound the gradient noise term as $4|\langle AB, G \rangle| \le 8\lambda \Vert A\Vert _F^2 + \frac{\Vert G(t)\Vert_2^2}{2\lambda} \Vert B\Vert_F^2$. Dropping the non-positive term, we obtain a clean linear differential inequality:
>
> $$v'(t) \le \frac{\Vert G(t)\Vert _2^2}{2\lambda} v(t) + 8\lambda r.$$
>
> Taking expectations under the variance assumption $\mathbb{E}[\Vert G(t)\Vert_2^2] \le \mu^2 + \sigma^2$ and applying Grönwall's inequality, we obtain the tightened expected energy bound over a training horizon $T$ by using Fubini's theorem:
>
> $$\sup_{t \in [0, T]} \mathbb{E}[v(t)] \le V_{max} := v_0 e^{\gamma T} + \frac{16\lambda^2 r}{\mu^2 + \sigma^2} \left( e^{\gamma T} - 1 \right),$$
>
> where $\gamma = \frac{\mu^2 + \sigma^2}{2\lambda}$ is the expected exponential growth rate.
>
> ## b. Refining Theorem 2: Rigorous Riccati Stability Analysis
>
> In the original manuscript, we omitted the positive higher-order term $+8\lambda d^2(t)$ as an simplified processing to highlight the convergence behavior near the origin. Retaining this nonlinear term allows for a strict global Riccati stability analysis, and the following analysis will reveal that our original asymptotic bound still holds without any approximations.
>
> We resume from the exact Riccati differential inequality:
>
> $$d'(t) \le 8\lambda d^2(t) - 8\lambda d(t) + K(t),$$
>
> where $K(t) = 2 a(t)b(t)\Vert G(t)\Vert_2$. From Young' s inequality, $K(t) \leq \bar{K} = \sup_t \mathbb{E}[K(t)] \le V_{max} \sqrt{\mu^2 + \sigma^2}$. The characteristic function $F(d)$ has roots $r_{1,2} = \frac{1}{2} \left(1 \pm \sqrt{1 - \frac{\bar{K}}{2\lambda}}\right)$.
>
> For the system to admit real equilibrium roots (i.e., for the optimization to stabilize rather than diverge), the discriminant must be non-negative, requiring $\lambda \ge \frac{\bar{K}}{2}$. This theoretically justifies that the orthogonal regularization strength $\lambda$ cannot be arbitrarily small.
>
> Using an auxiliary exact scalar differential equation and the Comparison Principle, since standard near-orthogonal initialization gives $d(0) \approx 0 < r_2$, the system globally converges to the stable attractor $r_1$:
>
> $$\limsup_{t \to \infty} d(t) \le r_1 = \frac{1}{2} \left( 1 - \sqrt{1 - \frac{\bar{K}}{2\lambda}} \right).$$
>
> In practical regimes where regularization dominates the gradient noise, the second-order Taylor expansion yields:
>
> $$r_1 \approx \frac{\bar{K}}{8\lambda} \le \frac{V_{max} \sqrt{\mu^2 + \sigma^2}}{8\lambda} \sim \mathcal{O}\left(\frac{\mu+\sigma}{\lambda}\right).$$
>
> We will fully incorporate these formalized derivations into the revised Appendix A. We thank the reviewer again for elevating the theoretical standard of our work.
>
> ---
>
> # 4. Difficulty in tuning hyperparameters
>
> In our follow-up experiments, we further found that fixing $\lambda=0.05$ already works well once the model size exceeds 1B. We therefore view the observation as suggesting that orthogonal regularization may become easier to tune, and potentially more beneficial, at larger model scales. And we attribute this phenomenon to the redundancy in larger LLMs.

---

> > ### Author Rebuttal · Reviewer_BWFH · 2026-04-03
> >
> > I thank the authors for providing a strong rebuttal.
> >
> > My concerns were on the following:
> >
> > 1) Lack of a proper ablation study - which the authors provided (albeit to another reviewer but noted here).
> >
> > 2) Comparison on other benchmarks, where LaVA performs roughly on par with DoRA.
> >
> > 3) Better theoretical justification - provided by the authors in the rebuttal.
> >
> > Overall, my main concerns have been addressed and I am increasing my score to Weak Accept.

---

> > > ### Author Response · Authors · 2026-04-03
> > >
> > > Thank you for the thoughtful and encouraging review. We greatly appreciate your positive assessment of our paper.

---

### Official Review · Reviewer_R6yf · 2026-03-12

**Soundness:** 3
**Presentation:** 3
**Significance:** 2
**Originality:** 3
**Overall Recommendation:** 4
**Confidence:** 4

**Summary:**

This paper tries to address some issues in the current implementation of LoRAs, namely the inefficient use of lower rank subspaces for model finetuning and the spatial disruption effect of flattening convolution operators for LoRA training. To solve these problems they introduce orthogonal regularization on the factor matrices to reduce column correlation and ensure better use of the subspace, and CP-decomposition that reparametrizes an incremental weight update as a summation of rank-1 tensors. The authors claim that this finetuning approach respects the multi-dimensional geometry of the weight tensors. They then test this approach for multiple settings involving both vision and language tasks under a unified implementation framework. This includes RoBERTa on GLUE, LLaMA2 on commonsense reasoning tasks, SAM for segmentation, Depth-Anything for depth estimation, and SDXL for text-to-image generation. Across these experiments, the authors report consistent but relatively small improvements over LoRA and several other PEFT methods.

**Compliance With Llm Reviewing Policy:**

Affirmed.

**Final Justification:**

The authors present a simple and interesting approach to PEFT. The empirical comparison is broad (and results have been further strengthened in the rebuttal), but empirical gains are relatively modest and statistical significance is not fully clear.

**Key Questions For Authors:**

1. Why would simple flattening cause dimension disorder and disrupt properties? A lot of other architectures also rely on flattening, is there any previous work on this or a mathematical/theoretical justification or do you mean that this is about this particular approach alone?

2. I could have missed this but was there any ablation studies on how CP-decomposition and orthogonal regularization individually affect performance aside from the toy example?

**Limitations:**

Yes

**Strengths And Weaknesses:**

Strengths:

1. The authors present a single unified approach to PEFT for any problem setting. The approach is simple and interpretable and can be easily adapted to existing approaches.

2. The idea of representing weight updates as sums of rank-1 tensor components is closely related to classical tensor decomposition techniques and provides a more natural way to handle convolutions without reshaping them

3. The empirical study is broad, including multiple architectures and tasks such as NLP, vision, and diffusion models. This helps verify their performance claims and justify the use of a single PEFT framework for all

Weaknesses:

1. The claim that the flattening operations cause dimension disorder and special disruption is not convincingly backed up by systematic arguments. Flattening the convolution kernel for parameterization does not change the behavior of the convolution operation itself or any update step. It only rearranges the entries without destroying the underlying linear mapping. The actual spatial inductive bias comes from the convolutional computation. The claim that this “degrades the model’s performance” is not backed by a systematic analysis. It is equally plausible that the drop in performance is due to the suboptimal low rank approximation of the flattened matrix, not from the reshaping itself. It could also be that the reshape operator was not correctly used as dimensions are shuffled meaning the torch permute operator must first be used to correctly flatten the values. Finally the comparison itself isn’t entirely fair as the two approaches are fundamentally different. Any performance might come from the alternative approach rather than the “reshape free” property. So while the authors’ approach performs better, I don’t agree with the conclusion that the flattening operator itself breaks spatial encoding.

2. The authors mention other papers that use orthogonal weight parametrization but never compare to them. Furthermore, the comparison to other advanced LoRA methods does not seem complete (e.g. with those focused on convolution, such as LoRA-C and Conv-Adapter).

3. The method adds computational overhead and the gains are not necessarily very significant with some results having high overlap in the variance ranges (a significance test here would be beneficial)

---

> ### Author Rebuttal · Authors · 2026-03-30
>
> We sincerely appreciate the effort you’ve dedicated to providing constructive and insightful comments! Below, I will try to respond to your questions and give my understanding.
>
> ---
>
> # 1. Clarification on Flattening vs. Low-Rank Parameterization
>
> We agree that flattening itself is a bijective re-indexing operation and does not alter the convolution operator. Actually, our intended claim is narrower: when a convolutional kernel is first unfolded into a matrix and then finetuned by a low-rank update, the parameterization becomes unfolding-dependent and no longer explicitly preserves the kernel’s original multi-way structure. We provide comparisons across different unfolding choices in the following table so that the discussion is not tied to one specific reshape order (all variants are compared under the same hyperparameters):
>
> | Model     | Out\*(in\*h\*w) | h\*(Out\*in*w) | w\*(out\*in\*h) | reshape-free |
> | --------- | --------------- | -------------- | --------------- | ------------ |
> | ResNet18  | 76.16%          | 77.69%         | 80.56%          | 92.37%       |
> | ResNet34  | 74.87%          | 80.40%         | 82.19%          | 93.85%       |
> | VGG11     | 64.36%          | 81.31%         | 82.50%          | 90.09%       |
> | ConvNeXt  | 76.09%          | 88.01%         | 88.32%          | 98%          |
> | GoogLeNet | 79.44%          | 83.47%         | 81.64%          | 93.52%       |
>
> Across all valid unfolding choices we tested, unfolded-matrix low-rank parameterization consistently underperforms the reshape-free tensor-form update. We therefore revise our claim as follows: unfolded-matrix low-rank parameterization can introduce a structural bias that is less suitable for convolutional tensors than a direct tensor-form update. We also believe that finetuning modules in tensor form helps discover the hidden connections better. Such hypotheses can be supported by recent work [1].
>
> ---
>
> # 2. Clean ablation study
>
> We agree that a cleaner ablation study is needed to isolate the influences of the two components. Below, we provide an ablation study when finetuning stable-diffusion-XL on the dataset pokemon-blip-capations. In the two ablation studies, all the hyperparameters are kept the same except (1): modules to be finetuned; (2): whether to apply orthogonal regularization or not.
>
> ## Panel A. Attention-only (matrix setting)
>
> | Setting | Attention adapter | Conv Adapter | Orthogonal regularization | Metric $\uparrow$ |
> | :-----: | :---------------: | :----------: | :-----------------------: | ----------------- |
> |   A1    |       LoRA        |      -       |           False           | 29.28             |
> |   A2    |       LoRA        |      -       |           True            | 29.42             |
>
> ## Panel B. Full U-Net (unified setting)
>
> | Setting | Attention Adapter |        Conv Adapter        | orthogonal regularization | Metric $\uparrow$ |
> | :-----: | :---------------: | :------------------------: | :-----------------------: | ----------------- |
> |   B1    |       LoRA        | Reshape-involved low-rank  |           False           | 28.10             |
> |   B2    |       LoRA        | Reshape-free decomposition |           False           | 29.62             |
> |   B3    |       LoRA        | Reshape-free decomposition |        True (Conv)        | 29.64             |
> |   B4    |       LoRA        | Reshape-free decomposition |        True (both)        | 29.96             |
>
> ---
>
> Based on the ablation above, we can get the following conclusions:
>
> 1.   Orthogonal regularization helps finetuning both matrix form (A1 vs. A2) and convolution form (B2 vs. B3).
> 2.   Low-rank parameterization is not suitable for the unfolded matrix (A1 vs. B1).
> 3.   Although convolution layer serves as feature extraction in vision tasks (normally treated as frozen when well pre-trained), finetuning the pre-trained convolution layer helps (A1 vs. B2).
> 4.   Finetuning convolutional layers in tensor form benefits (B1 vs. B2).
>
> \[1]: Zheng et al., ReFTA: Breaking the Weight Reconstruction Bottleneck in Tensorized Parameter-Efficient Fine-Tuning, in CVPR 2026.

---

> > ### Author Rebuttal · Reviewer_R6yf · 2026-04-02
> >
> > Thank you for the effort and clarification.
> >
> > The ablation experiments, additional benchmarks for reviewer bwfh, and the context for reshaping are definitely helpful and improve the method’s claim.
> >
> > However, my concerns about the statistical significance are not addressed as the improvements are still relatively modest.
> >
> > Overall, the rebuttal does make me feel more positive about the paper and I will adjust my score accordingly.

---

> > > ### Author Response · Authors · 2026-04-03
> > >
> > > Thank you for the thoughtful and encouraging review. We greatly appreciate your positive assessment of our paper.

---

### Official Review · Reviewer_CbPW · 2026-03-13

**Soundness:** 3
**Presentation:** 3
**Significance:** 3
**Originality:** 3
**Overall Recommendation:** 4
**Confidence:** 3

**Summary:**

This paper introduces LAVA, a unified framework for parameter-efficient fine-tuning (PEFT) that combines both convolution and attention modules, addressing limitations of LoRA. LAVA leverages low-rank subspaces and orthogonal regularization to improve training stability and enhance learning capabilities. The paper provides empirical validation of LAVA's effectiveness across NLP and vision tasks, demonstrating significant performance improvements over LoRA and other PEFT methods.

**Compliance With Llm Reviewing Policy:**

Affirmed.

**Final Justification:**

The authors addressed most of my concerns in the rebuttal. I will keep my score.

**Key Questions For Authors:**

See above.

**Limitations:**

Yes.

**Strengths And Weaknesses:**

### Strengths
- The paper introduces LAVA, a method that builds upon the well-established concepts of low-rank adaptation and orthogonal regularization. The theoretical framework behind LAVA is sound, with clear justifications for its design choices.
- The paper is well-structured, with a logical flow that clearly explains the problem, the proposed solution, and the experimental setup. The results are presented clearly, and the comparison with existing methods is understandable.


### Weaknesses
- The core components of LAVA, including tensor decomposition and orthogonal regularization, are not entirely novel and have been explored in previous works. A more detailed discussion of related techniques would help contextualize the contribution. The lack of comparison with these methods diminishes the originality of the work.
- LAVA has been tested across NLP and vision tasks. Can the authors provide insight into its applicability to other modalities, such as speech recognition or time-series forecasting? How does LAVA perform on tasks where both sequential and spatial data are involved?
- In addition to orthogonal regularization, could other forms of regularization be incorporated into LAVA to enhance its performance in specific domains or tasks? What impact would combining multiple regularization techniques have on the overall model performance?
- Does LAVA's low-rank approach help improve generalization compared to standard fine-tuning methods, especially on unseen or out-of-domain data?

---

> ### Author Rebuttal · Authors · 2026-03-30
>
> We greatly appreciate the reviewer’s careful reading and constructive feedback. Below, we try to address the questions and clarify the relevant points.
>
> # 1. Comparisons with related works
>
> ## a. Orthogonal matrix in PEFT methods
>
> HRA [1] bridges low-rank and orthogonal adaptation using householder reflections; SORSA [2] combines SVD-based adaptation with orthonormal regularization and analyzes its effect on conditioning. However, LAVA incorporates orthogonality into a low-rank **tensor-form** adaptation to preserve the simplicity of additive PEFT.
>
> ## b. Tensorized PEFT methods
>
> ReFTA [3] argues that existing PEFT methods are limited by layer-wise low-rank structure and proposes to stack modules and finetune them in a tensor form; TLoRA [4] uses tensor-train matrix to form rotation matrix, and tensor-ring decomposition for the residual. ReFTA focuses on using the relevance across layers and TLoRA studies the low-rank approximation gap. Instead, LAVA focuses on reshape-free tensor-form update for convolutional kernels, while still supporting original matrix tuning within a unified framework.
>
> ## c. PEFT for convolutional layers
>
> Conv-LoRA [5] injects convolutional modules into LoRA to adapt Segment Anything model; Conv-Adapter [6] studies efficient transfer for ConvNets using lightweight adapter modules. In contrast, LAVA does not add extra convolutional branches or adapter blocks and can be applied beyond models from vision fields.
>
> ---
>
> # 2. LAVA's performances in other modalities
>
> We have tested it in speech modality against other PEFT methods, and the results show that LAVA is good enough to beat other methods. We chose to finetune the pre-trained model Wav2Vec2 (1B) on dataset superb_ks. The results are shown in the following table:
>
> | Method |  ACC   |
> | :----: | :----: |
> |  LoRA  | 98.51% |
> |  DoRA  | 98.56% |
> |  LAVA  | 98.63% |
>
> We believe LAVA has shown great potential in other modalities as well.
>
> ---
>
> # 3. Incorporations of other forms of regularizations
>
> We have tried to apply component-wise dropout (Because of the rank-1 form of LAVA, here we only consider component-wise regularizations) and reported the performance in the following table:
>
> | Regularization form | Accuracy |
> | :-----------------: | :------: |
> |        Ortho        |  98.63%  |
> |       Dropout       |  98.54%  |
> |   Ortho + Dropout   |  98.60%  |
>
>
>
> These results suggest that component-wise dropout is a reasonable alternative, but it does not provide a consistent improvement over orthogonal regularization in our current setting. We therefore keep orthogonal regularization as the main choice in the current submission, since it remains the most reliable option among the regularizers we tested.
>
> There are many other regularization forms deserve to be tested: SoRA [7] introduces rank-wise gates to encourage sparsity; Weight normalization \[8] suggests separating magnitude and directions apart and then introducing a learnable scale to control the magnitude of a specific component; For dense prediction tasks, we can further use TV regularization [9]. These are all tightly aligned with the structure of LAVA rather than general element-wise penalties. Due to time constraints, we will extend them to future work.
>
> ---
>
> # 4. Generalization performances in OOD tasks
>
> In the commonsense reasoning experiment, the base model is finetuned on a dataset named commonsense_170k.json from LLM-Adapters [10] and then tested on eight datasets with different distributions from training set. Since LAVA performs better than other baselines in the commonsense reasoning task, we think that it is an indicator of better generalization performances in OOD setting.
>
> \[1]: Yuan et al., Bridging The Gap between Low-rank and Orthogonal Adaptation via Householder Reflection Adaptation, in NeurIPS 2024.
>
> \[2]: Cao et al., SORSA: Singular Values and Orthonormal Regularized Singular Vectors Adaptation of Large Language Models, arXiv: 2409.00055.
>
> \[3]: Zheng et al., ReFTA: Breaking the Weight Reconstruction Bottleneck in Tensorized Parameter-Efficient Fine-Tuning, in CVPR 2026.
>
> \[4]: Tao et al., Transformed Low-rank Adaptation via Tensor Decomposition and Its Applications to Text-to-image Models, arXiv: 2501.08727.
>
> \[5]: Zhong et al., CONVOLUTION MEETS LORA: PARAMETER EFFICIENT FINETUNING FOR SEGMENT ANYTHING MODEL, in ICLR 2024.
>
> \[6]: Chen et al., Conv-Adapter: Exploring Parameter Efficient Transfer Learning for ConvNets, in CVPR Workshop 2024.
>
> \[7]: Ding et al., Sparse Low-rank Adaptation of Pre-trained Language Models, in EMNLP 2023.
>
> \[8]: Salimans et al., Weight Normalization: A Simple Reparameterization to Accelerate Training of Deep Neural Networks, in NeurIPS 2016.
>
> \[9]: Osher et al., AN ITERATIVE REGULARIZATION METHOD FOR TOTAL VARIATION-BASED IMAGE RESTORATION, in Multiscale Modeling & Simulation 2005.
>
> \[10]: Hu et al., LLM-Adapters: An Adapter Family for Parameter-Efficient Fine-Tuning of Large Language Models, in EMNLP 2023.

---

> > ### Author Rebuttal · Reviewer_CbPW · 2026-04-04
> >
> > Thank you for the rebuttal. The new experimental results are helpful and improve the paper. I will retain my score.

---

> > > ### Author Response · Authors · 2026-04-06
> > >
> > > Thank you for the thoughtful and encouraging review. We greatly appreciate your positive assessment of our paper.

---

### Decision · Program_Chairs · 2026-04-30

**Decision:**

Accept (regular)

**Comment:**

This paper introduces LAVA (Language And Vision Adaptation), a parameter-efficient fine-tuning framework designed to adapt both attention-based and convolution-based modules across NLP and vision tasks. Instead of reshaping tensors to 2D matrices, the authors use a sum of outer products of vectors that are initialized to be orthogonal, and also add an orthogonalization term during optimization. The motivation for this work is that LoRA tends to underperform compared to its available capacity; e.g. the effective rank of the learned weights tends to be much smaller than the rank made available during optimization. Experimentally, LAVA performs better than previous variants of LoRA, across several tasks such as Natural Language Understanding, Semantic Segmentation, Depth Estimation, and Text-to-Image Generation. They also include ablations in their rebuttal.

Overall, the paper is well-structured and motivated and the experimental results are strong. Given the unanimous positive reviews, I recommend acceptance. I encourage the authors to address the comments raised by the reviewers in their camera-ready version (especially the ablations) and to pay attention to typos (e.g. there are typos in the abstract itself).